# LEARNING EFFECTIVE MULTI-MODAL TRACKERS VIA MODALITY-SENSITIVE TUNING

## ABSTRACT

This paper tackles the critical issue of constructing multi-modal trackers by effectively adapting the extensive knowledge of pre-trained RGB trackers to auxiliary modalities. To address the challenges, we propose a novel modality sensitivity-aware tuning framework, namely *MST*, which delicately models the learning process via adaptive tuning of model weights by inherent modality characteristics. Specifically, we first investigate the parameter modality-sensitivity as a criterion for measuring a precise element-wise essentiality for multi-modal adaptation. Then, in the tuning phase, we further leverage such sensitivity to bolster the stability and coherence of multi-modal representations, thereby enhancing generalization capabilities. Extensive experiments showcase the effectiveness of the proposed method, surpassing current state-of-the-art techniques across various multi-modal tracking scenarios and demonstrating remarkable performance even in extreme conditions. *The source code will be publicly available.*

## 1 INTRODUCTION

Object tracking, a foundation task of visual perception, has seen significant advancements over the past decades (Hong et al., 2024; Xie et al., 2024; Zheng et al., 2024; Cai et al., 2024). Despite the promising results, RGB-based trackers often struggle with some complex and degraded conditions, such as extreme illumination, motion blur, and occlusions. Therefore, multi-modal tracking with more comprehensive information (e.g., event, depth, thermal) has garnered growing interest. With the popularity of the data-driven methods in the object tracking community, both data scale and model size have got huge explosions (Ye et al., 2022; Lin et al., 2022; Chen et al., 2023). There is a prevailing paradigm that explores pre-trained trackers on large-scale RGB-based datasets and adapts them to diverse auxiliary modalities, a process known as cross-modal fine-tuning or transfer learning, to enhance performance and accelerate convergence.

Some existing approaches follow the *full fine-tuning* (FFT) paradigm (Tang et al., 2022; Wang et al., 2023; Zhu et al., 2023c), where the models are initialized with pretrained weights and are tuned by elaborately designed task-specific objectives. This type of method investigates cross-modal alignment to enhance the connection among modalities and obtain compact multi-modal representations. Nevertheless, due to the significant distribution gap and limited scale of auxiliary modalities, they are intractable to retain the pre-trained knowledge in the transfer phase and tend to induce catastrophic forgetting and over-fitting. In contrast to full fine-tuning, recent research has shifted toward *parameter efficient fine-tuning* (PEFT) (Jia et al., 2022; Chen et al., 2022; Lian et al., 2022). The core principle of PEFT is to keep the majority of pre-trained parameters frozen, updating or introducing only a small fraction of task-specific parameters to preserve pretrained prior knowledge. Several methods fall under this umbrella (Yang et al., 2022;

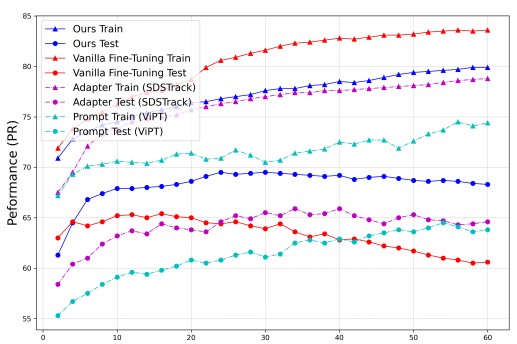

Figure 1: **The training and testing performance on the LasHeR dataset over the training phase, between our method and existing FFT and PEFT methods. Our method effectively mitigates the ill-fitting problem, and enhances the stability and generalization of the multi-modal tracker.**

Zhu et al., 2023a; Hou et al., 2024), including prompt tuning, visual adapter, etc., aiming at directly shifting and scaling to modulate the pre-trained patterns. Prompt tuning adapts the features of pre-trained vision transformers by introducing trainable auxiliary-modal tokens into one or more attention layers. Visual adapters insert some lightweight, nonlinear adapters to adjust for cross-modal distribution shifts. While effective and powerful, PEFT methods impose strong constraints on the primary model weight, resulting in a limited capacity for handling the vast distribution drift caused by different modalities.

This work seeks to refine the fine-tuning process to mitigate the ill-fitting issue (over- or under-fitting) for cross-modal tracking. To this end, we propose a modality sensitivity-aware framework that minimizes empirical risk while modulating parameter updates, thereby smoothing the adaptation process to jointly optimize both modal-specific and modal-agnostic general representations. Specifically, we optimize the learning dynamics of cross-modal trackers from the following two key perspectives. **(a)** *Modeling Parameter-wise Modality Sensitivity*. To learn robust representation from multi-modal data, we first leverage the training objective as a criterion to reflect the influence of parameters' variations, facing the multi-modal data, which is so-called multi-modal sensitivity. Moreover, we approximate such multi-modal sensitivity via the off-the-shelf gradient matrix from the training process. **(b)** *Modality Sensitivity-Aware Adaptive Tuning*. With the aforementioned modality sensitivity, we then construct an adaptive tuning scheme. It preserves the prior knowledge from pretrained model via adaptively adjusting the learning step according to the precise modality sensitivity. By incorporating such a modality sensitivity-aware regularization of parameter learning dynamics, our approach effectively preserves pre-trained knowledge and facilitates seamless transfer to multi-modal tracking tasks, which can continuously enhance the model in the training phase (please refer to Fig. 1).

Our method strategically guides the cross-modal fine-tuning process to optimize downstream tasks while preserving the generalization capacity of the pre-trained model. Extensive experimental results showcase our method achieves new state-of-the-art results on all benchmarks. Comprehensive ablation studies demonstrate the effectiveness of the self-regularized fine-tuning concept.

In summary, the main contributions of this paper are:

- we revisit the ill-fitting issue of cross-modal tracking for adapting foundational models and propose a self-regularized fine-tuning framework to indicate better generalization, which is in contrast to the existing FFT and PEFT methods;

- we propose to exploit the parameter modality sensitivity to regularize the parameter updating and suggest a self-ensemble weight strategy over iterations to enhance the stability and consistency of multi-modal representations, thereby facilitating the generalization; and

- we conduct comprehensive experiments covering three multi-modal tracking tasks and five datasets and push cross-modal tracking accuracy to new levels.

## 2 RELATED WORK

**Multi-Modal Tracking.** Object tracking is one of the cornerstone tasks in computer vision, involving predicting the position and scale of an object in subsequent frames given an arbitrary object in the initial frame. In recent years, due to the unreliability of RGB-only data in challenging scenarios, increasing studies are expected to integrate auxiliary modalities (e.g., event, depth, thermal) to enhance tracking performance. For example, (Zhang et al., 2021a; 2023; 2024a) combine RGB and event streams to predict objects in low-dynamic scenarios. (Mueller et al., 2017; Liu et al., 2018; Qian et al., 2021) incorporate additional depth maps for tracking in occlusion environments. Similarly, (Wang et al., 2020; Zhang et al., 2021b; Hui et al., 2023) fuse thermal infrared data to obtain reliable appearance and motion cues. In summary, these delicate and impressive methods focus on effective feature interaction and fusion across multiple modalities (Tang et al., 2022; Zhang et al., 2023; Wang et al., 2023; Zhang et al., 2024a). With the emergence of large-scale datasets and universal backbones (e.g., vision transformer), pre-trained trackers (Ye et al., 2022; Wu et al., 2023) have demonstrated remarkable generalization capabilities. As a result, there is a growing tendency to explore pre-trained models in multiple auxiliary modalities to further enhance performance. In this work, we focus on optimizing the use of pre-trained knowledge for efficient cross-modal transfer.

**Cross-modal Transfer Learning.** To facilitate training multi-modal trackers with the pre-trained ones, two primary types of efforts have recently been made. Some works follow the full fine-tuning (FFT) paradigm (Tang et al., 2022; Wang et al., 2023; Zhu et al., 2023c), which draw upon the pre-trained models for initialization and update it by elaborately designing cross-modal alignment objectives. These methods desire a shared/compact feature space to inherit the generalization capability of the original model, by aligning auxiliary modal with RGB. Although effective, one primary drawback may be innate to this paradigm: the overfitting is significant due to the contradiction between the paucity of large-scale auxiliary datasets and the huge appetites of the cross-modal transfer process. Profited by the affluent experience of natural language processing (NLP) and computer vision (CV) communities (e.g., prompt tuning (Jia et al., 2022), visual adapter (Chen et al., 2022), and LoRA (Hu et al., 2021), etc), some parameter efficient fine-tuning (PEFT) methods (Yang et al., 2022; Zhu et al., 2023a; Cao et al., 2024; Hou et al., 2024) have been proposed. These techniques involve tuning only a minimal number of additional parameters for downstream tasks while keeping the pre-trained weights frozen. For instance, ViPT modulates the RGB features by introducing trainable auxiliary-modal tokens into multiple attention layers. SDSTrack inserts a lightweight module with a bottleneck architecture between attention layers to address cross-modal distribution shifts. Despite their effectiveness, these modulated models tend to overfit since the additional parameters are optimized from scratch with respect to the modal-specific objective. Furthermore, these PEFT methods impose strong constraints on the pre-trained weights, often incurring insufficient transfer learning. Thus, how to efficiently fine-tune the RGB-based pre-trained models for target modalities remains a major challenge.

# 3 PROPOSED METHOD

Learning robust and effective multi-modal latent representations is crucial for adapting RGB-based neural models to multi-modal tasks. To fully boost the capabilities of multi-modal object tracking, we re-examine the fundamentals of multi-modal adaptation and object tracking. For clarity, we first illustrate the architectural designs, which mainly consist of cascaded self-attention layers in Sec. 3.1. We then investigate to quantify the crucial modality sensitivity in Sec. 3.2. Finally, in Sec. 3.3, such sensitivity is designed to integrate into the tuning phase of multi-modal tracking, which dynamically panels the parameter-wise updation with an accumulated scheme.

## 3.1 OVERVIEW OF NETWORK ARCHITECTURE

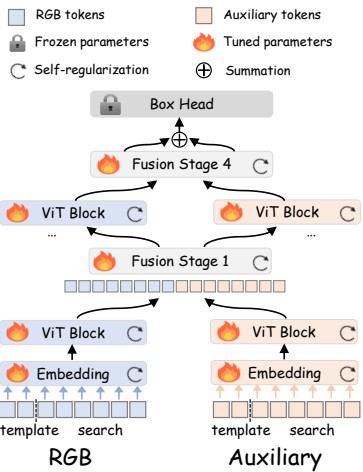

Figure 2: Overview of Architecture.

Fig. 2 depicts the overall architecture of our method. The RGB and auxiliary inputs are first fed to the embedding layer to generate the corresponding tokens. Then the symmetric transformer backbones (ViT) are used for feature extraction and interaction. Without involving customized multi-modal fusion modules, we reuse part of the ViT blocks to implement multi-stage fusion among multi-modal tokens. To retain the modal-agnostic object association knowledge, we utilize and freeze the pre-trained box head. Last, we take the fusion features as the input for the head. **In contrast** to existing *full fine-tuning* and the *parameter-efficient fine-tuning* paradigms, which usually result in over-fitting and under-fitting, respectively, we seek to refine the fine-tuning process to address the issue of ill-fitting. To this end, we propose a self-regularizing method that guides the training process to efficiently transfer the generalization of RGB-based pre-trained trackers to auxiliary modalities and fuses multi-modal features effectively.

Denote by $f_{\boldsymbol{\theta}}(\cdot)$ a multi-modal tracker, where $\boldsymbol{\theta} = \{\boldsymbol{\theta}_1, \cdots, \boldsymbol{\theta}_N\}$ the corresponding parameters of the multi-modal tracker with a total number of $N$ parameter. Consider a training set $\mathbf{D} = \{(x_i, y_i) | i = 1, \cdots, M\}$ with $M$ total samples, where $x_i$ is a multi-modal data pair, and $y_i$ is the corresponding bounding-box label. Multi-modal tracking aims to learn a well-generalized model by fine-tuning the $\boldsymbol{\theta}$. The vanilla fine-tuning strategy first applies the pre-trained parameter

to initialize it and update it based on the objective:

$$\boldsymbol{\theta}^{(i+1)} = \boldsymbol{\theta}^{(i)} - \alpha \nabla_{\boldsymbol{\theta}^{(i)}} \mathcal{L}\left(\boldsymbol{\theta}^{(i)}; \mathbf{D}\right) \quad s.t. \quad \boldsymbol{\theta}^{(0)} := \boldsymbol{\theta}^{(p)}, \tag{1}$$

where $\mathcal{L}(\cdot)$ is the loss function, $\alpha$ is the learning rate, $i$ indicates the iteration step, and $\boldsymbol{\theta}^{(p)}$ represents weights of the pre-trained neural network.

## 3.2 Modeling Modality Sensitivity for Multi-Modal Tracking

To learn a robust and effective multi-modal tracker from pre-trained weights, the critical issue lies in the correct modeling of gradient descent direction. Moreover, by assuming that a robust and effective neural network should equivalently and effectively leverage all parameters, in this section, we first investigate the parameter sensitivity for the multi-modal task, which quantifies the contributions of each parameter for neural network performance.

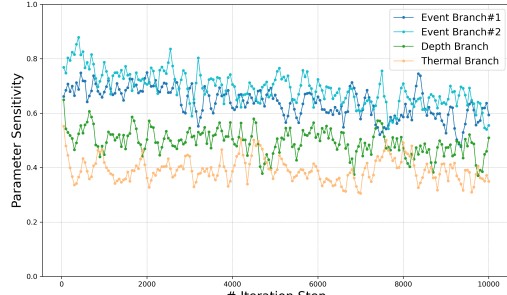

Figure 3: **The parameter-wise sensitivity across different auxiliary modalities. For a given parameter, similarities occur within the modality and differences across modalities during training. Event Branch#1 and #2 correspond to the VisEvent and CoeSot datasets, respectively.**

To measure the algorithm's performance, we utilize the value of learning objectives, which is easily derived. Thus, the influence of parameters' variations can be quantified by the following general formulation,

$$\mathcal{S}_{\boldsymbol{\theta}}(\boldsymbol{\epsilon}) = \mathcal{L}\left(\boldsymbol{\theta}; \mathbf{D}\right) - \mathcal{L}\left(\boldsymbol{\theta}'; \mathbf{D} \mid \boldsymbol{\theta}' = \boldsymbol{\theta} + \boldsymbol{\epsilon}\right), \tag{2}$$

where $\mathcal{L}(\cdot) : \mathbb{R}^n \to \mathbb{R}^1$ indicates the inference process, which maps the network weights with training samples to the value of training objective; $\boldsymbol{\theta} \in \mathbb{R}^n$ indicates the weights of pretrained model; $\boldsymbol{\theta}'$ represents the corresponding weights of $\boldsymbol{\theta}$ perturbed by a small noise $\boldsymbol{\epsilon} \in \mathbb{R}^n$; and $\mathcal{S}(\cdot) : \mathbb{R}^n \to \mathbb{R}^1$ represents the neural network sensitive function, which shows the variations of neural network performance given the parameter perturbation. Through expanding $\mathcal{L}\left(\boldsymbol{\theta}'; \mathbf{D}\right)$ as the Taylor series over $\boldsymbol{\theta}$ and omitting the high-order terms, we can derive $\mathcal{S}(\boldsymbol{\epsilon}) \approx \frac{\partial \mathcal{L}}{\partial \boldsymbol{\theta}} \boldsymbol{\epsilon}^T$.

As the aforementioned, an effective method should ensure that the network does not focus on specific parameters, and different perturbations should result in similar responses. We then start to find an optimal weight $\boldsymbol{\theta}^*$ by

$$\boldsymbol{\theta}^* = \underset{\boldsymbol{\epsilon}, \boldsymbol{\epsilon}' \sim \mathbf{P}_n}{\arg\min} \|\mathcal{S}_{\boldsymbol{\theta}}(\boldsymbol{\epsilon}) - \mathcal{S}_{\boldsymbol{\theta}}(\boldsymbol{\epsilon}')\|_2, \tag{3}$$

where $\boldsymbol{\epsilon}$ and $\boldsymbol{\epsilon}'$ are two different noise samples from the same distribution $\mathbf{P}_n$. Moreover, under the assumption that the total energy of gradient matrix is fixed, i.e., $\|\frac{\partial \mathcal{L}}{\partial \boldsymbol{\theta}}\|_2 = C$ ($C$ denotes a scalar constant), we can easily derive a closed-form solution of equation 3, that all elements from gradient vector $\frac{\partial \mathcal{L}}{\partial \boldsymbol{\theta}}$ have the same magnitude, i.e., $\frac{\partial \mathcal{L}}{\partial \boldsymbol{\theta}_n} = \pm \frac{C}{\sqrt{N}}, n = 1, \cdots, N$.

We bring the general formulation of equation 2 to the dynamic learning process of equation 1. Here, we adjust the parameter $\boldsymbol{\theta}$ via the gradient $\nabla_{\boldsymbol{\theta}} \mathcal{L}\left(\boldsymbol{\theta}; \mathbf{D}\right)$[1] instead of previously mentioned $\boldsymbol{\epsilon}$. Thus, we can easily derive $\mathcal{S}(\nabla_{\boldsymbol{\theta}} \mathcal{L}\left(\boldsymbol{\theta}; \mathbf{D}\right)) = \frac{\partial \mathcal{L}}{\partial \boldsymbol{\theta}} \left[\nabla_{\boldsymbol{\theta}} \mathcal{L}\left(\boldsymbol{\theta}; \mathbf{D}\right)\right]^T = \left\langle \frac{\partial \mathcal{L}}{\partial \boldsymbol{\theta}}, \frac{\partial \mathcal{L}}{\partial \boldsymbol{\theta}} \right\rangle$, where $\langle \cdot, \cdot \rangle$ indicates the inner product of two vectors. Note that aforementioned sensitivity $\mathcal{S}(\nabla_{\boldsymbol{\theta}} \mathcal{L}\left(\boldsymbol{\theta}; \mathbf{D}\right))$ is designed for analysis the adjustment for all the parameters in $\boldsymbol{\theta}$. To delicately investigate and regularize the gradient, we further extend aforementioned sensitivity into a parameter-wise formulation as $s_n = \frac{\partial \mathcal{L}}{\partial \boldsymbol{\theta}_n}^2$, where $s_n$ denote the sensitivity of the $n^{th}$ parameter.

Based on the above analysis, we can summarize that for a robust and effective multi-modal tracker, the gradient matrix should be uniformly distributed. It indicates neural networks equivalently leverage different parameters. To utilize such characteristics to boost the training of cross-modal trackers, we illustrate the method that integrates the parameter-wise sensitivity $s_n$ into the learning process of cross-modal trackers in the next section.

---

[1] Both terms $\nabla_{\boldsymbol{\theta}} \mathcal{L}\left(\boldsymbol{\theta}; \mathbf{D}\right)$ and $\frac{\partial \mathcal{L}}{\partial \boldsymbol{\theta}}$ represent the same meaning of gradients from the loss function $\mathcal{L}$ on the parameter $\boldsymbol{\theta}$. However, they are introduced for different utilization, i.e., $\nabla_{\boldsymbol{\theta}} \mathcal{L}\left(\boldsymbol{\theta}; \mathbf{D}\right)$ for network optimization and $\frac{\partial \mathcal{L}}{\partial \boldsymbol{\theta}}$ for sensitivity analysis.

### 3.3 Modality Sensitivity-Aware Tuning of Multi-modal Trackers

We regularize the learning process by previously discussed modality-sensitivity to derive modality-robust trackers. However, the established formulations are based on the gradient from the whole dataset, i.e., gradient descent with full dataset $\mathbf{D}$. It makes a naive implementation to be computationally expansive and even intractable. Thus, in this section, we further explore the temporal correlation of the learning dynamics to naturally combine the sensitivity-aware tuning into the learning process of multi-modal trackers.

---

**Algorithm 1:** Modality Sensitivity-aware Tuning

---

**Input:** Pre-trained model $\boldsymbol{\theta}^{(0)}$,
   initialized momentum $\boldsymbol{\rho}^{(0)}$,
   training set $\mathbf{D}$

**Output:** Optimal parameters $\boldsymbol{\theta}^*$

**for** $i \in \{1, \ldots, K\}$ **do**
 | Get the $i$-th batch data $\mathbf{M}_i$ from $\mathbf{D}$;
 | Compute loss $\mathcal{L}$ and gradients $\mathbf{G}^{(i)}$;
 | Update sensitivity by
 | $\mathcal{S}^{(i)} = \beta \left\langle \frac{\partial \mathcal{L}}{\partial \boldsymbol{\theta}^{(i)}}, \frac{\partial \mathcal{L}}{\partial \boldsymbol{\theta}^{(i)}} \right\rangle$;
 | Momentum update by Eq. (4);
 | Parameter update via Eq. (5);
**end**

---

Since the full-dataset gradient is an expectation over $\mathbf{D}$, we establish a memory-capable gradient to further adapt the introduced modality sensitivity tuning strategy. Specifically, we deploy a momentum-driven gradient to modulate the parameter update, where the sensitivity serves as the momentum coefficient. Formally, we compute the gradient w.r.t. $\mathbf{G}^{(i)} := \partial \mathcal{L}\left(\boldsymbol{\theta}; \mathbf{M}_i\right) / \partial \boldsymbol{\theta}$ and the sensitivity $s_n^{(i)}$, where the $i$-th iteration of parameter update is illustrated:

$$\boldsymbol{\rho}^{(i+1)} = \mathcal{S}^{(i)} \odot \boldsymbol{\rho}^{(i)} + \left(1 - \mathcal{S}^{(i)}\right) \odot \mathbf{G}^{(i)} \quad (4)$$

$$\boldsymbol{\theta}^{(i+1)} = \boldsymbol{\theta}^{(i)} - \alpha \boldsymbol{\rho}^{(i+1)}, \quad (5)$$

where $\mathbf{M}_i$ represents a mini-batch, $\boldsymbol{\rho}^{(i)}$ denotes the momentum gradient with our modality sensitivity ($\boldsymbol{\rho}^{(0)} = \mathbf{0}$), $\mathcal{S}^{(i)}$ indicates the previously mentioned sensitivity, shown in Algorithm 1 with $\beta$ for a re-scaling factor, $\odot$ for Hadamard product, and $\alpha$ is the learning rate. To adapt to the momentum updates (Sutskever et al., 2013), we rank the sensitivity metrics and apply a linear mapping to a continuous range $[a, b]$, as the subset of $[0, 1]$. (Please refer to Sec 4.3 for more analysis). The momentum update in Eq. (4) suggests that more sensitive parameters should retain their previous states to a greater extent, to avoid oscillations or over-adjustments. This manner allows $\boldsymbol{\theta}$ to evolve more smoothly than its vanilla counterpart. As a result, the tracker for the different samples is kept as consistent as possible despite its evolution.

**Remark.** As analyzed, most gradient-aware sensitivity methods, such as (Zhang et al., 2024b; He et al., 2023; Fu et al., 2023), select the most sensitive parameters for sparse fine-tuning. In contrast, our method prioritizes penalizing these sensitive parameters. Furthermore, we avoid masking to forcefully restrict the parameter solution space. For temporal-aware weight aggregation, some methods ensemble pre-trained or cross-epoch weights (Wortsman et al., 2022; Khattak et al., 2023). Instead, our approach applies iteration-level parameter modulation. We also discuss these potentially viable tuning methods in the *Appendix A.3*.

### 3.4 Learning Objectives

The overall loss function of ours is the same as the foundation model without extra adjusting, shown as:

$$\mathcal{L} = \mathcal{L}_{\text{cls}} + \lambda_{\text{iou}} \mathcal{L}_{\text{iou}} + \lambda_{l_1} \mathcal{L}_1, \quad (6)$$

where $\mathcal{L}_{cls}$ is the weighted focal loss for classification, $l_1$ loss $\mathcal{L}_1$ and GIoU loss $\mathcal{L}_{iou}$ are employed for bounding box regression, $\lambda_{iou} = 2$ and $\lambda_{l_1} = 5$ are the regularization factors, and all the corresponding settings are the same as (Ye et al., 2022).

## 4 Experiments

### 4.1 Experimental Settings

**Datasets and Metrics.** To verify the effectiveness and generalization of the proposed method, we conduct comprehensive experiments on multiple multi-modal benchmark datasets. Our tracker is evaluated on FE108 (Zhang et al., 2021a), VisEvent (Wang et al., 2023) and CoeSot (Tang et al., 2022) for RGB-Event tracking, DepthTrack (Yan et al., 2021b) for RGB-Depth tracking, and LasHeR (Li et al., 2020) for RGB-Thermal tracking. For object tracking, we utilize four widely

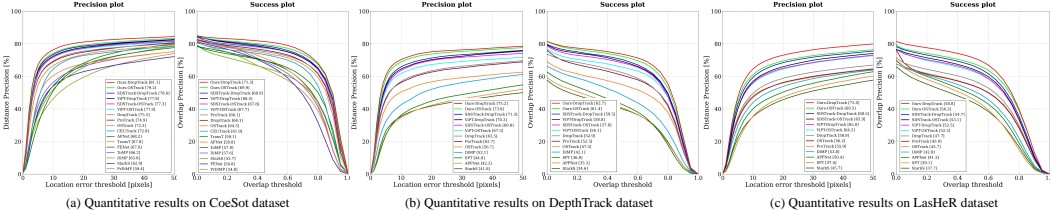

Figure 4: Visualization of the precision and success plots of the CoeSot, DepthTrack, and LasHeR datasets. We also refer readers to the *Appendix A.3* for more comprehensive evaluations. Zoom in to see details.

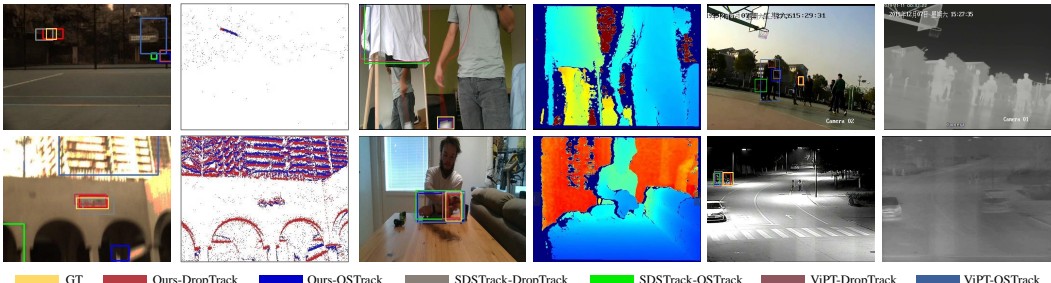

Figure 5: Visual comparisons of the tracking performance of different methods on the (**Left**) CoeSot, (**Middle**) DepthTrack and (**Right**) LasHeR datasets.

used metrics for comparisons, i.e., representative success rate (RSR), representative precision rate (RPR), and overlap precision (OP) with the threshold equal to 0.5 ($OP_{0.5}$) and 0.75 ($OP_{0.75}$). **For DepthTrack benchmark, we use precision (Pr) and recall (Re) to measure the performance. F-score, calculated by $F = \frac{2RePr}{Re+Pr}$, is its primary measure.**

**Pre-trained Models and Baselines.** In this paper, we choose two classic one-stream RGB-based trackers, e.g., OSTrack (Ye et al., 2022) and DropTrack (Wu et al., 2023), as the pre-trained models. Notably, these two trackers adopt the ViT-B/16 (Dosovitskiy et al., 2020) as the backbone. Corresponding to the pre-trained settings, we present two variants with different input resolution: OSTrack (Template: 128×128, Search: 256×256); DropTrack (Template: 192×192, Search: 384×384). To objectively and sufficiently validate the effectiveness of our method, we conduct the following experiments. First, we construct several top-notch RGB-based trackers as single-modal baselines, which follow the full fine-tuning fashion. Then, we compare our method with a variety of cross-modal transfer protocols, including the full fine-tuning and the parameter-efficient fine-tuning paradigms. Moreover, we extend the current state-of-the-art trackers to their DropTrack versions. Notably, the methods with pre-trained OSTrack and DropTrack are compared separately.

**Training Details.** We follow the data processing of SDSTrack (Hou et al., 2024) in all the datasets. The models are trained on 8 NVIDIA 3090Ti GPUs with a batch size of 192 and 30 epochs. Each epoch involves sampling 80k samples. We utilize the AdamW optimizer with a learning rate set to $1 \times 10^{-4}$ and a weight decay set to $10^{-4}$.

### 4.2 COMPARISON WITH STATE-OF-THE-ART METHODS

Extensive comparative analyses are presented in Tab. 1 and Tab. 2, where our method demonstrates excellent performance on all multi-modal benchmarks after applying the proposed techniques during training. The corresponding precision and success plots are shown in Fig. 4. Evidently, we can observe that both the RGB-only and the cross-modal strategies are becoming increasingly profitable with pre-trained models. In particular, cross-modal approaches exhibit substantial performance gains, highlighting the complementarity between RGB and auxiliary data under complex conditions. Importantly, the notable improvement of our method suggests the significance and necessity of pursuing suitable cross-domain generalization methods for multi-modal object tracking.

Table 1: Quantitative comparison on the three RGB-Event dataset (FE108, VisEvent and CoeSot). For all metrics, the **larger**, the **better**. The best results are marked with **"bold"**, and the second best results are marked with "underline". "†" indicates our implementation.

| Method | Base Model | FE108 | | | | VisEvent | | | | CoeSot | | | |
|---|---|---|---|---|---|---|---|---|---|---|---|---|---|
| | | RSR | $OP_{0.5}$ | $OP_{0.75}$ | RPR | RSR | $OP_{0.5}$ | $OP_{0.75}$ | RPR | RSR | $OP_{0.5}$ | $OP_{0.75}$ | RPR |
| **Image-based Methods (Only-RGB)** | | | | | | | | | | | | | |
| DiMP (Bhat et al., 2019) ICCV'19 | ResNet | 48.5 | 60.2 | 22.3 | 72.6 | 50.3 | 60.4 | 42.2 | 66.3 | 57.6 | 66.9 | 48.9 | 65.6 |
| PrDiMP (Danelljan et al., 2020) CVPR'20 | ResNet | 47.5 | 57.9 | 20.6 | 72.5 | 46.8 | 56.1 | 39.1 | 60.7 | 54.8 | 62.4 | 45.8 | 59.4 |
| TransT (Chen et al., 2021) CVPR'21 | ResNet | 47.9 | 57.5 | 21.6 | 72.8 | 45.1 | 53.9 | 38.9 | 58.6 | 59.1 | 68.4 | 55.8 | 67.8 |
| Stark-S (Yan et al., 2021a) ICCV'21 | ResNet | 50.6 | 61.0 | 23.9 | 76.0 | 41.3 | 48.8 | 34.5 | 53.7 | 55.7 | 63.9 | 50.5 | 62.9 |
| ToMP (Mayer et al., 2022) CVPR'22 | ResNet | 50.1 | 61.2 | 21.8 | 76.1 | 38.3 | 45.7 | 31.3 | 50.4 | 57.9 | 68.1 | 51.6 | 66.2 |
| OSTrack (Ye et al., 2022) ECCV'22 | OSTrack | 48.5 | 59.8 | 26.3 | 70.7 | 55.9 | 66.4 | 52.7 | 69.5 | 64.3 | 73.2 | 64.4 | 73.3 |
| DropTrack (Wu et al., 2023) CVPR'23 | DropTrack | 52.0 | 65.1 | 29.7 | 74.9 | 57.1 | 68.1 | 54.1 | 71.3 | 66.1 | 75.3 | 66.2 | 75.5 |
| **Cross-modal Transfer Learning** | | | | | | | | | | | | | |
| FENet (Zhang et al., 2021a) ICCV'21 | DiMP | 61.6 | 78.7 | 34.7 | 91.0 | 51.3 | 61.6 | 42.2 | 67.9 | 57.8 | 68.3 | 55.5 | 69.8 |
| AFNet (Zhang et al., 2023) CVPR'23 | DiMP | 61.5 | 80.3 | 31.1 | 90.9 | 51.1 | 61.3 | 42.5 | 67.5 | 59.6 | 69.7 | 54.3 | 69.6 |
| CEUTrack (Tang et al., 2022) - | OSTrack | 55.6 | 73.2 | 30.4 | 84.5 | 56.2 | 66.8 | 53.2 | 69.9 | 61.9 | 72.4 | 61.3 | 72.8 |
| ProTrack (Yang et al., 2022) MM'22 | OSTrack | 58.0 | 74.6 | 30.0 | 84.5 | 57.1 | 67.6 | 53.2 | 71.6 | 66.1 | 74.6 | 66.3 | 74.9 |
| ViPT (Zhu et al., 2023a) CVPR'23 | OSTrack | 64.9 | 84.2 | 39.9 | 92.4 | 59.5 | 70.5 | 57.6 | 73.8 | 67.7 | 76.8 | 68.5 | 77.0 |
| SDSTrack (Hou et al., 2024) CVPR'24 | OSTrack | 60.0 | 77.1 | 34.4 | 86.1 | 59.8 | 71.0 | 57.9 | 74.2 | 67.8 | 77.0 | 68.5 | 77.3 |
| ViPT (Zhu et al., 2023a) CVPR'23 | DropTrack | 65.1 | 84.0 | 41.7 | 91.9 | 60.4 | 72.1 | 57.5 | 75.0 | 68.5 | 77.2 | 68.7 | 77.8 |
| SDSTrack (Hou et al., 2024) CVPR'24 | DropTrack | 65.6 | 84.5 | 42.2 | 93.4 | 61.5 | 73.4 | 59.1 | 76.4 | 68.9 | 78.1 | 69.4 | 78.6 |
| **Ours†** | OSTrack | 67.4 | 87.1 | 44.2 | 95.5 | 62.1 | 73.8 | 61.0 | 76.5 | 69.9 | 79.0 | 71.2 | 79.2 |
| **Improvement** | OSTrack | +2.5 | +2.9 | +4.3 | +3.1 | +2.3 | +2.8 | +3.1 | +2.3 | +2.1 | +2.0 | +2.7 | +1.9 |
| **Ours†** | DropTrack | 68.7 | 89.2 | 47.1 | 96.0 | 63.2 | 75.4 | 61.4 | 78.1 | 71.3 | 80.9 | 72.4 | 81.1 |
| **Improvement** | DropTrack | +3.1 | +4.7 | +4.9 | +2.6 | +1.7 | +2.0 | +2.3 | +1.7 | +2.4 | +2.8 | +3.0 | +2.5 |

**Results on RGB-Event.** As illustrated in Tab. 1, our method surpasses all state-of-the-art trackers across all RGB-Event datasets, achieving the highest precision of 96.0%, 78.1% and 81.1% on the FE108, VisEvent, and CoeSot datasets, respectively. Notably, on FE108, our method exceeds the previous top results by a large extent: 3.1% in RSR, 4.7% in $OP_{0.5}$, 4.9% in $OP_{0.75}$, and 2.6% in RPR. full fine-tuning approaches such as CEUTrack fall short in limited improvement. Contrarily, parameter efficient fine-tuning paradigms like ViPT and SDSTrack attain remarkably competitive results. However, these methods encounter performance bottlenecks on FE108, which relies on event data and includes extensive low-light scenes, likely due to architectural modifications that adversely affect the cross-modal transfer potentiality.

**Results on RGB-Depth.** As shown in the left of Tab. 2, our method outperforms all previous state-of-the-art trackers on the DepthTrack, obtaining the top performance of 75.2% and 62.7% in precision and success, significantly exceeding prior best results. Using the pre-trained OSTrack, our method yields substantial improvements: 4.2% in Pr, 4.1% in Re, and 4.1% in F-score. Similarly, based on the pre-trained DropTrack, our method shows significant gains: 4.3% in Pr, 3.9% in Re, and 4.1% in F-score.

**Results on RGB-Thermal.** As listed in right of Tab. 2, our method surpasses all previous state-of-the-art trackers on the LasHeR, achieving the new state-of-the-art performance of 73.0% and 58.8% in precision and success, which exceeds the SDSTrack by a significant margin, i.e., 4.1% in RSR, 4.9% in $OP_{0.5}$, 4.8% in $OP_{0.75}$, and 4.5% in RPR. Importantly, our method further unleashes the potential of the pre-training model with more knowledge (i.e., DropTrack), and yields a greater performance gain. We reason such an effect may result from promoting thoroughly cross-modal transfer learning.

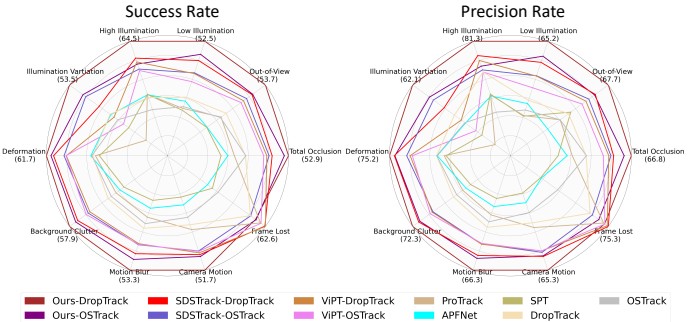

Figure 6: Attribute analysis on LasHeR.

Table 2: Quantitative comparison on the two reflective RGB-Depth and RGB-Thermal datasets (DepthTrack and LasHeR). The best results are marked with **"bold"**, and the second/ best results are marked with "underline"."†" indicates our implementation.

| Method | Base Model | DepthTrack | | | | LasHeR | | |
|---|---|---|---|---|---|---|---|---|
| | | Pr | Re | F-score | RSR | $OP_{0.5}$ | $OP_{0.75}$ | RPR |
| Image-based Methods (Only-RGB) | | | | | | | | |
| DiMP (Bhat et al., 2019) ICCV'19 | ResNet | 46.3 | 42.8 | 44.5 | 42.8 | 51.3 | 30.3 | 53.8 |
| Stark-S (Yan et al., 2021a) ICCV'21 | ResNet | 39.3 | 37.6 | 38.4 | 37.7 | 44.1 | 23.6 | 45.7 |
| OSTrack (Ye et al., 2022) ECCV'22 | OSTrack | 53.6 | 52.2 | 52.9 | 45.7 | 54.6 | 36.7 | 56.2 |
| DropTrack (Wu et al., 2023) CVPR'23 | DropTrack | 56.4 | 55.8 | 56.1 | 47.7 | 57.7 | 38.4 | 58.8 |
| Cross-modal Transfer Learning | | | | | | | | |
| SPT (Zhu et al., 2023b) AAAI'23 | Stark-S | 52.7 | 54.9 | 53.8 | 39.1 | 46.1 | 22.6 | 47.4 |
| APFNet (Xiao et al., 2022) AAAI'22 | Stark-S | 51.6 | 51.4 | 51.5 | 41.3 | 48.7 | 27.1 | 50.4 |
| ProTrack (Yang et al., 2022) MM'22 | OSTrack | 58.3 | 57.3 | 57.8 | 45.9 | 54.3 | 36.3 | 55.9 |
| ViPT (Zhu et al., 2023a) CVPR'23 | OSTrack | 59.2 | 59.6 | 59.4 | 52.5 | 63.1 | 43.2 | 64.5 |
| SDSTrack (Hou et al., 2024) CVPR'24 | OSTrack | 61.9 | 60.9 | 61.4 | 53.1 | 64.5 | 43.6 | 65.9 |
| ViPT (Zhu et al., 2023a) CVPR'23 | DropTrack | 62.6 | 61.6 | 62.1 | 52.5 | 63.7 | 42.9 | 65.0 |
| SDSTrack (Hou et al., 2024) CVPR'24 | DropTrack | 63.5 | 62.4 | 62.9 | 54.7 | 66.6 | 45.4 | 68.5 |
| **Ours**[†] | OSTrack | **66.1** | **65.0** | **65.5** | **56.3** | **68.0** | **48.5** | **69.5** |
| **Improvement** | OSTrack | +4.2 | +4.1 | +4.1 | +3.2 | +3.5 | +4.9 | +3.6 |
| **Ours**[†] | DropTrack | **67.8** | **66.3** | **67.0** | **58.8** | **71.5** | **50.2** | **73.0** |
| **Improvement** | DropTrack | +4.3 | +3.9 | +4.1 | +4.1 | +4.9 | +4.8 | +4.5 |

**More Comparisons and Analyses.** In addition, we perform analysis of various challenging attributes, such as illumination variation, motion blur, out-of-view, etc. As shown in Fig. 6, it can be seen that we also achieve the best tracking performance in these extreme scenarios. For example, the proposed regularization achieves 7.2% precision and 5.3% success improvements under the low illumination attribute. We also refer readers to *Appendix A.2* for a more detailed attribute analysis.

## 4.3 ABLATION STUDY AND ANALYSIS

**Effectiveness of Proposed Components.** We conduct comprehensive experiments to better understand the relationship and effectiveness of the two proposed regularization technique, as shown in Tab 3. Comparing **(a)** and **(b)** demonstrats that fine-tuning significantly enhances the domain adaptation ability. Further, to mitigate the over-fitting issue in fine-tuning, we propose two regularization technologies. As shown in Tab 3, comparisons between **(b)** and **(c)** (or **(d)** and **(e)**) figure out that the sensitivity-aware scheme significantly improves the RSR by 2.1% and RPR by 3.0% on the LasHeR dataset, highlighting its effectiveness. Moreover, comparing **(b)** and **(d)** (or **(c)** and **(e)**) shows that interpolating parameter dynamics optimized with different data leads to substantial improvements. Notably, it can be seen that applying both techniques simultaneously yields much more significant improvements than using either method alone. These observations confirm that the two techniques are complementary. Although the improvement from sensitivity regularization alone is not substantial on the FE108, it still contributes to the method's leading performance.

Table 3: Ablative study results of the proposed key components. Note that all methods are based on the pre-trained OSTrack. **"F-Tune."** refers to fully fine-tuning the model, where we only re-train the backbone; **"Param.Sens."** represents constraining parameter updates based on their sensitivities; and **"Momen."** indicates to interpolate parameters from successive iterations. **(a)** denotes the zero-shot performance. **(b)** serves as our baselines,

| Exp. | F-Tune. | Param. Sens. | Momen. | FE108 | | | | DepthTrack | | | LasHeR | | | |
|---|---|---|---|---|---|---|---|---|---|---|---|---|---|---|
| | | | | RSR | $OP_{0.5}$ | $OP_{0.75}$ | RPR | Pr | Re | F-score | RSR | $OP_{0.5}$ | $OP_{0.75}$ | RPR |
| (a) | | | | 48.1 | 59.5 | 16.6 | 74.2 | 38.2 | 36.0 | 37.1 | 36.7 | 41.2 | 26.8 | 42.9 |
| (b) | ✓ | | | 65.2 | 84.3 | 41.9 | 93.0 | 61.7 | 61.5 | 61.6 | 53.2 | 64.1 | 45.3 | 65.4 |
| (c) | ✓ | ✓ | | 66.5 | 85.8 | 43.7 | 94.1 | 63.4 | 62.5 | 62.9 | 55.3 | 67.0 | 47.2 | 68.4 |
| (d) | ✓ | | ✓ | 66.6 | 85.3 | 42.9 | 94.3 | 64.1 | 63.1 | 63.6 | 55.0 | 66.3 | 46.9 | 67.9 |
| (e) | ✓ | ✓ | ✓ | 67.4 | 87.1 | 44.2 | 95.5 | 66.1 | 65.0 | 65.5 | 56.3 | 68.0 | 48.5 | 69.5 |

**Effectiveness on Single-modal Methods.** This work aims to mitigate the overfitting issue when adapting the foundation models to downstream tasks. A key question is how the proposed regularization methods perform on single-modal data. To investigate this, we conduct ablation studies

highlighting their impact on different modalities. As shown in Tab 4, both RGB and Auxiliary modalities benefit significantly from the proposed regularization techniques. For example, the RGB and Depth gain $5.1\%$ F-score improvements on the DepthTrack dataset. Notably, despite huge distribution differences, our method significantly and consistently enhances the adaptability of Auxiliary modal across multiple tasks. These findings underscore the importance and necessity of imposing constraints when transferring the pre-trained trackers to downstream data.

Table 4: Ablation results of the proposed regularization fine-tuning method on single-modal data. **"RGB/Auxiliary"** refers to the fine-tuning fashion. Note that all methods are built upon the pre-trained OSTrack.

| Exp. | CoeSot | | | | DepthTrack | | | LasHeR | | | |
|------|--------|--------|--------|-----|-----|-----|---------|-----|--------|--------|-----|
| | RSR | $OP_{0.5}$ | $OP_{0.75}$ | RPR | Pr | Re | F-score | RSR | $OP_{0.5}$ | $OP_{0.75}$ | RPR |
| RGB | 64.3 | 73.2 | 64.4 | 73.3 | 53.9 | 53.0 | 53.4 | 47.2 | 56.4 | 37.6 | 58.0 |
| RGB+Ours | 68.0 | 76.8 | 69.2 | 76.9 | 58.7 | 58.4 | 58.5 | 50.3 | 60.2 | 41.2 | 62.0 |
| **Improvement** | **+3.7** | **+3.6** | **+4.8** | **+3.6** | **+4.8** | **+5.4** | **+5.1** | **+3.1** | **+3.8** | **+3.6** | **+4.0** |
| Auxiliary | 57.7 | 67.6 | 52.8 | 67.2 | 49.1 | 47.4 | 48.2 | 42.7 | 51.8 | 29.3 | 53.1 |
| Auxiliary+Ours | 60.3 | 70.8 | 55.2 | 70.4 | 52.6 | 51.2 | 51.9 | 45.8 | 55.2 | 32.6 | 57.0 |
| **Improvement** | **+2.6** | **+3.2** | **+2.4** | **+3.2** | **+3.5** | **+3.8** | **+3.7** | **+3.1** | **+3.4** | **+2.9** | **+3.9** |

**Compatibility with PEFT Methods.** Existing PEFT methods (Yang et al., 2022; Zhu et al., 2023a; Hou et al., 2024) typically freeze pre-trained parameters and update only a minimal number of additional parameters, which may limit sufficient optimization. To assess the compatibility of our proposed regularization techniques with existing PEFT methods, we unfreeze their backbone parameters and retrain with regularization applied (detailes in *Appendix A.1*). As shown in Tab 5, for ViPT, our method yields notable gains on LasHeR: $2.4\%$ in RSR and $3.3\%$ in RPR. These results affirm that overly constraining pre-trained models limit their transfer potential. However, our method negatively impacts the performance of SDSTrack. This occurs because our method optimizes the pre-trained parameters, whereas SDSTrack introduces modal-specific adapters learned from scratch.

Table 5: **Compatibility study results of the proposed regularization fine-tuning method on ViPT and SDSTrack, using the pre-trained OSTrack weights. "F-Tune." denotes full fine-tuning of the backbone, while "Self-Reg." represents the self-regularization scheme.**

| Exp. | VisEvent | | | | DepthTrack | | | LasHeR | | | |
|------|----------|--------|--------|-----|-----|-----|---------|-----|--------|--------|-----|
| | RSR | $OP_{0.5}$ | $OP_{0.75}$ | RPR | Pr | Re | F-score | RSR | $OP_{0.5}$ | $OP_{0.75}$ | RPR |
| ViPT | 59.5 | 70.5 | 57.6 | 73.8 | 59.2 | 59.6 | 59.4 | 52.5 | 63.1 | 43.2 | 64.5 |
| ViPT + **F-Tune.** | 57.5 | 68.4 | 54.4 | 72.0 | 58.2 | 57.4 | 57.8 | 50.9 | 61.7 | 41.6 | 63.2 |
| ViPT + **F-Tune.** + **Self-Reg.** | 61.4 | 72.9 | 59.8 | 75.8 | 61.7 | 61.1 | 61.4 | 54.9 | 66.1 | 46.2 | 67.8 |
| SDSTrack | 59.8 | 71.0 | 57.9 | 74.2 | 61.9 | 60.9 | 61.4 | 53.1 | 64.5 | 43.6 | 65.9 |
| SDSTrack + **F-Tune.** | 55.6 | 66.9 | 51.2 | 70.4 | 57.8 | 56.4 | 57.1 | 50.6 | 61.7 | 40.4 | 63.2 |
| SDSTrack + **F-Tune.** + **Self-Reg.** | 57.5 | 68.9 | 54.9 | 72.1 | 59.2 | 58.0 | 58.6 | 52.4 | 63.7 | 42.5 | 65.0 |
| Ours | 62.0 | 73.7 | 60.9 | 76.5 | 66.1 | 65.0 | 65.5 | 56.3 | 68.0 | 48.5 | 69.5 |

**Settings of Momentum Coefficient.** In this section, we conduct an experimental evaluation to assess the impact of varying values of the momentum coefficient. The results are presented in Tab. 6, where we systematically increase the coefficient from 0.5 to 0.95, while evaluating different momentum ranges derived from parameter sensitivity mapping. We observe that a moderately large momentum coefficient (e.g., 0.8) works better than both smaller (e.g., 0.5) and larger values (e.g., 0.95), suggesting that a relatively slow evolving encoder is key to effectively utilizing a data queue. When the parameter sensitivity is scaled to an appropriate range (e.g., $[0.8, 0.85]$), parameter updates are subject to dual constraints of sensitivity and momentum, thereby enhancing transfer capabilities.

Table 6: Ablation analysis of momentum coefficient on the FE108 dataset. Note that all methods are based on the pre-trained OSTrack weight. **"Mc."** refers to momentum coefficient; **Scalar** represents that all parameters employ the same momentum coefficient, while **Range** $[a, b]$ represents the momentum coefficients derived from parameter sensitivity mapping.

| Mc. | 0 | 0.5 | 0.6 | 0.7 | 0.75 | 0.8 | 0.85 | 0.9 | 0.95 | [0.75, 0.95] | [0.8, 0.9] | [0.85, 0.9] | [0.8, 0.85] |
|-----|---|-----|-----|-----|------|-----|------|-----|------|-------------|-----------|------------|------------|
| RSR | 65.2 | 65.6 | 65.7 | 66.2 | 66.4 | 66.6 | 66.5 | 66.3 | 65.8 | 66.9 | 67.1 | 67.1 | 67.4 |
| RPR | 93.2 | 93.2 | 93.4 | 93.8 | 94.0 | 94.3 | 94.4 | 93.9 | 93.6 | 94.8 | 95.1 | 95.2 | 95.5 |

**Observations on Sensitivity Patterns.** The sensitivity criterion identifies task-specific key patterns, highlighting the precedence of pre-trained parameters to downstream tasks. We visualize the sensi-

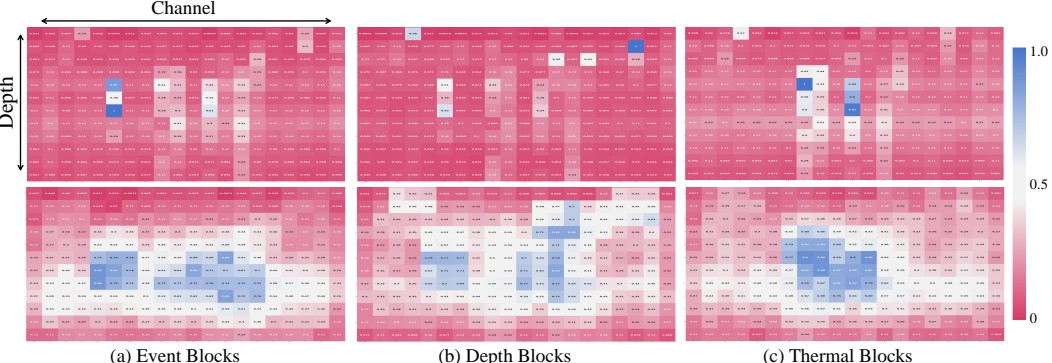

(a) Event Blocks         (b) Depth Blocks         (c) Thermal Blocks

Figure 7: Visualization of the parameter sensitivity patterns cross different auxiliary modalities (e.g., event, depth, thermal), where the colorbar denotes the sensitivity metric. The **upper** represents the pre-trained parameter patterns, while the **bottom** shows the parameter patterns well-tuned by our method.

tivity matrices of different auxiliary branches (ViT-B/16) in Fig. 7. We observed notable differences in sensitivity patterns, indicating a significant modal-aware bias. Additionally, the clustering of sensitive parameters in certain areas (**upper** of Fig. 7) indicates an over-reliance on specific parameters, hindering the global transfer of the pre-trained model. Following the application of sensitivity penalties, the patterns (**bottom** of Fig. 7) become more balanced and distributed, suggesting that mitigating parameter bias improves the model's generalization and robustness.

**Efficiency of Self-regularization Fine-tuning.** We compare the training efficiency of our method with vanilla fine-tuning. Our method introduces parameter-wise sensitivity quantification and momentum updates with marginal computational overhead, as sensitivity is derived from the off-the-shelf gradients. In vanilla fine-tuning, the training speed is 37.5 ms per iteration, while our method operates at 39.3 ms per iteration, resulting in only a $5.7\%$ increase in computational time.

**Computational Cost and Inference Speed.** Computational efficiency is a key consideration in object tracking. We compare the computational complexities and speeds. Note that the proposed regularization techniques are applied solely during training, imposing no additional computational burden during testing. **These experiments were run on the same computer with an Intel(R) Xeon(R) 6456C CPU, 256 GB RAM, and one NVIDIA 3090Ti GPU.** As shown in Tab. 7, our method enables real-time tracking at 29.2 frames per second, while delivering top-tier performance. Integrating our method, "ViPT+Ours" yields both superior speed and performance. Under the DropTrack base model, it still operates at 48.8 frames per second, maintaining a respectable speed given its focus on accuracy.

Table 7: Computational complexity and speed analysis on the LasHeR dataset. The marks "faster", "best" and "balance" signify the most superior speed, performance, and their optimal trade-off, respectively.

| Method | Base Model | Param (M) | Flops (G) | FPS | RSR | RPR |
|---|---|---|---|---|---|---|
| OSTrack | OSTrack | 92.1 | 58.1 | 98.4 | 45.7 | 56.2 |
| DropTrack | DropTrack | 92.1 | 130.6 | 57.6 | 47.7 | 58.8 |
| ProTrack | OSTrack | 92.7 | 58.4 | 92.3 | 45.9 | 55.9 |
| ViPT | OSTrack | 92.9 | 59.9 | 88.6 | 52.5 | 64.5 |
| ViPT | DropTrack | 92.9 | 131.9 | 48.8 | 52.5 | 65.0 |
| SDSTrack | OSTrack | 102.1 | 108.7 | 44.6 | 53.1 | 65.9 |
| SDSTrack | DropTrack | 102.1 | 244.2 | 26.8 | 54.7 | 68.5 |
| Ours | OSTrack | 202.0 | 149.0 | 49.0 | 55.9 | 69.2 |
| Ours | DropTrack | 202.0 | 335.3 | 29.2 | 58.3 | 72.5 |
| ViPT+Ours | OSTrack | 92.9 | 59.9 | 88.6 | 54.9 | 67.8 |
| ViPT+Ours | DropTrack | 92.9 | 131.9 | 48.8 | 57.1 | 70.5 |

## 5 CONCLUSION

This paper re-examined the critical issues of constructing multi-modal trackers by effectively transferring the extensive knowledge of pre-trained RGB trackers to auxiliary modalities. To this end, we introduced a novel modality sensitivity-aware tuning framework, by delicately modulating the learning process from two key perspectives. First, we leveraged the task objectives to reflect parameter sensitivity, enabling optimizing the updates of essential parameters. Further, in the intertemporal update, we deployed a momentum-driven gradient to bolster the stability and coherence of multi-modal representations. Extensive experiments demonstrate the effectiveness of the proposed method, surpassing current state-of-the-art techniques across various multi-modal tracking scenarios, with significant improvements observed post-regularization. We believe these insights will inspire further exploration of multi-modal transfer learning for scene perception.

## REPRODUCIBILITY STATEMENT

In Section 4.1 and Appendix A.1, we outline the configurations of the hyper-parameters, describe the training process, and detail the implementation aspects of our approach. We also offer a comprehensive explanation of the datasets used in our study. To ensure accuracy and reproducibility, we perform multiple experiments using the FE108 (Zhang et al., 2021a), VisEvent (Wang et al., 2023) and CoeSot (Tang et al., 2022), DepthTrack (Yan et al., 2021b), and LasHeR (Li et al., 2020) datasets. Furthermore, if our paper is accepted for publication at ICLR 2025, we will release the source code and configuration files on GitHub.

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

# A APPENDIX

This appendix contains the following contents. We illustrate the implementation details of MST in A.1, including the network architecture and training details. In A.2, we report more quantitative results, comprising the precision-success plots and attribute analysis. We also compare and discuss other potentially viable regularization methods in A.3. Moreover, we have supplemented some tracking visuals for a better qualitative comparison in A.4.

- Section A.1: Implementation details.
- Section A.2: More quantitative results.
- Section A.3: More ablation studies.
- Section A.4: Visualization of tracking results.

## A.1 IMPLEMENTATION DETAILS.

**Network Architecture.** The input of our proposed method consists of a pair of template frames and a pair of search frames, i.e., one RGB template frame $z^R \in \mathbb{R}^{H_z \times W_z \times 3}$, one RGB search frame $x^R \in \mathbb{R}^{H_x \times W_x \times 3}$, one Auxiliary-modal template frame $z^A \in \mathbb{R}^{H_z \times W_z \times 3}$, and one Auxiliary-modal search frame $x^A \in \mathbb{R}^{H_x \times W_x \times 3}$. **Notably, to make event data compatible with the RGB domain, we aggregate the event set between the image and its next one into a three-channel event frame.** These data are first split and flattened into sequences of patches $z_R, z_A \in \mathbb{R}^{N_z \times (3P^2)}$ and $x_R, x_A \in \mathbb{R}^{N_x \times (3P^2)}$, where $P \times P$ is the resolution of each patch, and $N_z = \frac{H_z W_z}{P^2}$, $N_x = \frac{H_x W_x}{P^2}$. Next, two modal-aware patch embedding layers are used to project $z_R, x_R$ and $z_A, x_A$ into the D-dimensional latent space, $z_R, z_A \in \mathbb{R}^{N_z \times D}$ and $x_R, x_A \in \mathbb{R}^{N_x \times D}$. The patch embeddings $z_R$ and $x_R$ are concatenated as $\mathbf{H}_R^{(0)} = [z_R; x_R] \in \mathbb{R}^{(N_z + N_x) \times D}$, and $z_A$ and $x_A$ are concatenated as $\mathbf{H}_A^{(0)} = [z_A; x_A] \in \mathbb{R}^{(N_z + N_x) \times D}$. The computation of modal-aware ViT block can be formulated as:

$$\mathbf{H}_X'^{(l)} = \mathbf{H}_X^{(l-1)} + \text{MSA}\left(\text{LN}\left(\mathbf{H}_X^{(l-1)}\right)\right)$$

$$\mathbf{H}_X^{(l)} = \mathbf{H}_X'^{(l)} + \text{MLP}\left(\text{LN}\left(\mathbf{H}_X'^{(l)}\right)\right)$$

where $X \in R, A$, $\mathbf{X}_X^{(l-1)}$ and $\mathbf{H}_X^{(l)}$ represent the outputs of the $(l-1)$-th and $l$-th ViT blocks, respectively. For the cross-modal block, we concatenate $\mathbf{H}_F = [z_R; x_R; z_A; x_A] \in \mathbb{R}^{(N_z + N_x + N_z + N_x) \times D}$ as input, and use the same attention block as above for cross-modal feature interaction.

**Evaluation Metrics.** Specifically, success rate cares the frame of that overlap between ground truth and predicted bounding box is larger than a threshold; We employ the area under curve (AUC) of a success rate plot as representative success rate (RSR). Precision rate focuses on the frame of that the center distance between ground truth and predicted bounding box within a given threshold; We use the precision rate score associated with a 20-pixel threshold as representative precision rate (RPR). $OP_T$ represents success rate with $T$ as the threshold, 0.5 ($OP_{0.5}$) and 0.75 ($OP_{0.75}$) represent the success rates under moderate and challenging conditions, respectively.

**Training Details.** Our method is evaluated on FE108 (Zhang et al., 2021a), VisEvent (Wang et al., 2023) and CoeSot (Tang et al., 2022) for RGB-Event tracking, DepthTrack (Yan et al., 2021b) for RGB-Depth tracking, and LasHeR (Li et al., 2020) for RGB-Thermal tracking. More precisely, the FE108 dataset is captured under different degraded conditions (e.g., motion blur, high dynamic range). We follow the official sequence splits: 76 sequences for training and 32 for testing. The VisEvent dataset reflects many challenging dynamic outdoor scenes like motion blur, fast and non-rigid motion, etc. **Notably, there are some sequences that miss $\star$.aedat file or have misaligned timestamps, the VisEvent dataset only includes 295 sequences for training and 219 for testing.** The CoeSot dataset consists of 578K image-event pairs, including 824 sequences for training and 528 for testing. DepthTrack is a large-scale long-term RGB-Depth tracking benchmark, which contains 152 training and 50 testing videos. LasHeR is a large-scale high-diversity benchmark for short-term RGB-Thermal tracking, it includes 979 sequences for training and 245 for testing. **For these datasets, there is a slight difference in the learning rate settings**. For the VisEvent, CoeSot, DepthTrack and LasHeR, we set the learning rate of RGB blocks and auxiliary-modal blocks

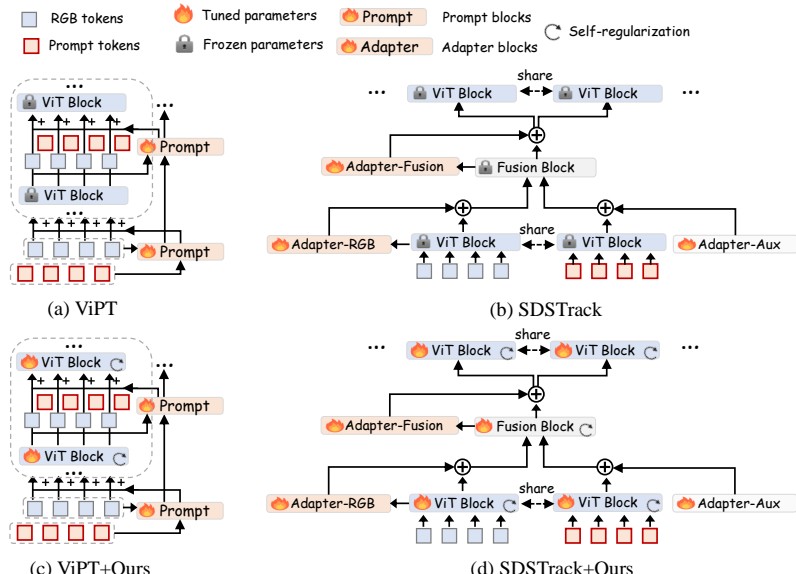

Figure 8: The detailed architecture of the proposed regularization fine-tuning method on the ViPT and SDSTrack.

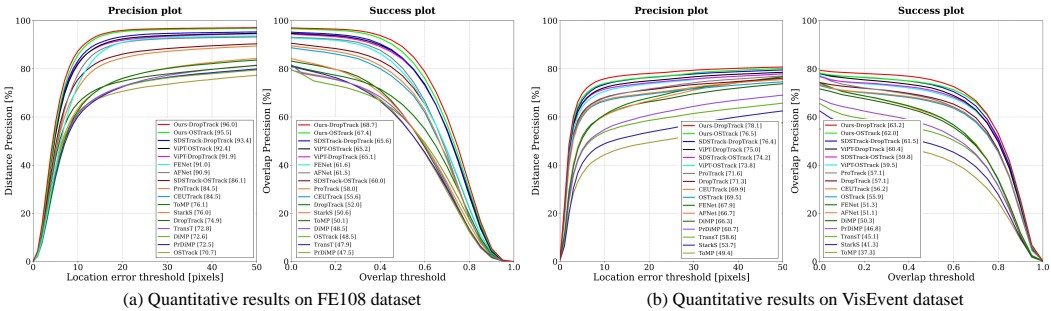

Figure 9: Visualization of the precision and success plots of the FE108 and VisEvent datasets.

to $10^{-4}$, and the learning rate of cross-modal blocks to $5 \times 10^{-5}$. But for the FE108 dataset, the learning rate of the RGB blocks, auxiliary-modal blocks and cross-modal blocks are set to $10^{-4}$, $5 \times 10^{-5}$ and $5 \times 10^{-5}$, respectively.

**Details for Compatibility Study.** To assess the compatibility of our proposed regularization techniques with ViPT and SDSTrack, we unfreeze their backbone parameters and retrain with regularization applied. For a clear understanding, please refer to Fig. 8.

## A.2 MORE QUANTITATIVE RESULTS.

**More Precision and Success Plots.** For a comprehensive and clear comparison, we also present the precision and success plots of FE108 and VisEvent datasets.

**Detailed Attribute Analysis.** In the VisEvent, CoeSot, and LasHe datasets, detailed attribute labels are available. To comprehensively analyze the robustness of our method, we compared its performance against previous methods on various challenging attributes. As shown in Tab 8 and Tab 9, Our method achieves state-of-the-art performance across most attributes in the VisEvent and CoeSot dataset. Specifically, in sequences involving motion, such as Camera Motion (CM), Background Object Motion (BOM), Fast Motion (FM), and Motion Blur (MB), our method consistently delivers the best results, highlighting its ability to accurately track objects despite degradation caused by movement. In sequences related to illumination, such as Low Illumination (LI), Over Exposure (OE), and

Abrupt Illumination Variation (AIV), our method demonstrates the best performance. Particularly, it achieves a precision improvement of $4.7\%$ and a success improvement of $6.6\%$ in the Over Exposure category, showcasing its adaptability to varying lighting conditions. For sequences involving occlusion, such as Partial Occlusion (PO) and Full Occlusion (FO), our method again achieves the best results, demonstrating its effectiveness in tracking targets even when partially or fully occluded. As shown in Tab 10, Our method is optimal on almost all attributes while significantly leading. Notably, our method outperforms other methods in sequences involving illumination interference, such as Low Illumination (LI), High Illumination (HI), and Abrupt Illumination Variation (AIV). Especially, it shows a precision rate of $81.3\%$ and a success rate of $64.5\%$ in High Illumination, a precision rate of $65.2\%$, and a success rate of $52.5\%$ in Low Illumination. This highlights its effective use of multi-modal information to enhance tracking robustness. Additionally, our method excels in challenging attributes such as Thermal Crossover, Background Clutter, Aspect Ratio Change, Full Occlusion, Out-of-View, Viewpoint Change, and Scale Variation, further showcasing its improved robustness. Overall, the results across the VisEvent, CoeSot, and LasHeR datasets demonstrate the strong performance and robustness of our method.

Table 8: Attribute performance on the **VisEvent** test set. The right superscript $^o$ represents the pre-trained OSTrack, and $^\dagger$ denotes the pre-trained DropTrack. The metric is the $RSR/RPR$.

| | OSTrack | DropTrack | FENet | AFNet | CEUTrack | ProTrack$^o$ | ViPT$^o$ | ViPT$^\dagger$ | SDSTrack$^o$ | SDSTrack$^\dagger$ | Ours$^o$ | Ours$^\dagger$ |
|---|---|---|---|---|---|---|---|---|---|---|---|---|
| Camera Motion | 56.0/69.3 | 58.5/72.7 | 39.2/51.6 | 37.5/51.0 | 56.3/69.8 | 56.3/70.5 | 58.1/77.2 | 61.2/76.1 | 58.7/73.0 | 61.8/76.8 | 61.5/75.8 | **63.6/78.5** |
| Rotation | 48.6/56.2 | 51.2/59.5 | 30.9/34.1 | 37.6/45.1 | 51.8/60.5 | 53.2/64.9 | 56.5/66.5 | 60.0/70.4 | 56.9/67.2 | **62.6/74.0** | 57.3/67.0 | 62.5/73.1 |
| Deformation | 35.3/41.8 | 36.1/44.2 | 28.0/35.3 | 26.4/33.3 | 34.6/41.0 | 29.5/37.8 | 33.3/41.6 | 36.3/45.3 | 39.0/48.1 | 40.5/52.3 | 37.2/45.0 | **42.1/53.1** |
| Full Occlusion | 42.9/55.9 | 48.7/62.3 | 19.1/29.0 | 24.4/35.0 | 46.0/59.1 | 46.2/59.6 | 499.4/63.0 | 53.3/68.0 | 50.1/63.9 | 55.0/69.9 | 52.8/66.9 | **56.0/70.8** |
| Low Illumination | 54.9/67.4 | 56.9/70.4 | 37.4/49.4 | 38.4/52.3 | 54.1/66.4 | 49.5/60.8 | 55.9/69.1 | 56.6/69.8 | 57.9/72.0 | 59.9/75.3 | 59.7/73.6 | **61.8/76.1** |
| Out-of-View | 43.3/53.2 | 47.8/59.1 | 21.7/28.1 | 27.6/39.6 | 45.8/56.7 | 43.1/52.3 | 45.8/55.9 | 51.2/62.3 | 47.1/58.0 | **52.7/64.4** | 59.4/60.5 | 52.6/64.4 |
| Partial Occlusion | 50.3/64.1 | 50.1/63.4 | 26.4/36.4 | 28.8/39.1 | 51.9/65.8 | 53.7/68.9 | 56.1/70.6 | 56.4/71.1 | 55.0/69.7 | 57.2/72.2 | 59.1/74.5 | **60.2/75.5** |
| Viewpoint Change | 68.9/69.2 | 61.3/72.8 | 40.7/47.6 | 44.2/53.0 | 61.2/71.8 | 60.1/71.4 | 63.1/73.9 | 64.6/76.0 | 63.4/74.3 | 65.7/78.0 | 67.4/79.8 | **68.6/81.2** |
| Scale Variation | 48.2/60.0 | 48.4/60.2 | 31.3/40.5 | 34.6/45.7 | 49.0/60.4 | 51.0/63.3 | 54.3/66.5 | 57.2/64.4 | 54.3/66.9 | 55.3/68.3 | 56.6/70.1 | **57.8/71.3** |
| Background Clutter | 54.7/67.9 | 55.1/68.8 | 36.5/48.4 | 35.5/48.1 | 54.5/67.8 | 56.3/71.0 | 58.6/72.7 | 58.4/72.5 | 58.0/71.9 | 59.5/73.8 | 60.4/74.6 | **61.3/75.7** |
| Motion Blur | 50.4/61.0 | 51.3/62.4 | 33.2/41.7 | 34.0/45.3 | 50.3/61.2 | 59.2/60.6 | 50.2/61.0 | 53.0/64.1 | 51.0/61.9 | **55.6/67.8** | 52.3/63.3 | 54.2/65.2 |
| Aspect Ration Change | 56.2/68.4 | 54.8/66.2 | 31.9/40.1 | 36.8/47.5 | 55.7/67.4 | 52.0/63.4 | 57.8/70.0 | 57.1/69.2 | 59.7/72.6 | 61.0/74.7 | 60.1/73.9 | **62.6/75.7** |
| Fast Motion | 51.3/63.0 | 53.9/66.4 | 34.3/42.9 | 37.3/49.6 | 53.0/65.2 | 52.2/64.7 | 54.1/66.4 | 57.2/70.0 | 54.8/67.3 | 59.1/72.4 | 57.0/69.3 | **59.4/72.4** |
| No Motion | 57.2/68.9 | 57.4/69.6 | 36.1/47.1 | 35.1/46.1 | 57.2/69.9 | 57.4/71.6 | 59.6/72.9 | 60.8/73.9 | 57.8/70.7 | 64.0/78.2 | 62.6/77.4 | **64.9/79.5** |
| Illumination Variation | 58.0/71.5 | 57.1/70.5 | 45.5/57.6 | 44.8/60.0 | 58.0/71.4 | 59.6/74.1 | 61.2/74.9 | 58.3/71.8 | 61.1/74.7 | 60.7/75.0 | **63.8/78.2** | 62.6/77.4 |
| Over Exposure | 54.9/69.9 | 53.5/66.3 | 43.0/55.3 | 43.3/54.8 | 54.1/68.1 | 59.1/74.3 | 58.8/74.0 | 55.5/69.1 | 59.1/73.6 | 56.6/70.0 | **63.2/78.3** | 59.8/75.5 |
| Background Object Motion | 53.0/66.2 | 54.0/67.5 | 34.2/45.1 | 33.6/45.4 | 53.1/66.3 | 55.0/69.9 | 56.9/71.0 | 57.6/71.9 | 56.5/70.3 | 58.7/73.2 | 59.1/73.2 | **60.4/74.9** |

Table 9: Attribute performance on the **CoeSot** test set. The right superscript $^o$ represents the pre-trained OSTrack, and $^\dagger$ denotes the pre-trained DropTrack. The metric is the $RSR/RPR$.

| | OSTrack | DropTrack | FENet | AFNet | CEUTrack | ProTrack$^o$ | ViPT$^o$ | ViPT$^\dagger$ | SDSTrack$^o$ | SDSTrack$^\dagger$ | Ours$^o$ | Ours$^\dagger$ |
|---|---|---|---|---|---|---|---|---|---|---|---|---|
| Camera Motion | 52.1/60.8 | 54.8/64.6 | 46.6/59.9 | 46.3/58.5 | 49.1/60.2 | 52.4/60.5 | 56.0/65.4 | 57.9/67.8 | 55.6/64.9 | 59.1/70.1 | 59.0/69.2 | **62.1/73.2** |
| Rotation | 71.9/81.8 | 72.4/82.6 | 56.2/67.6 | 63.0/72.2 | 71.0/83.0 | 72.1/81.1 | 74.1/'84.1 | 74.3/84.2 | 74.1/84.6 | 74.6/84.8 | 76.2/86.6 | **76.4/86.9** |
| Deformation | 67.1/68.5 | 72.7/75.9 | 56.7/62.2 | 60.0/63.5 | 71.6/80.1 | 75.0/80.7 | 77.3/83.0 | 75.1/79.6 | 75.7/81.4 | 75.3/79.7 | 77.1/81.8 | **75.8/81.0** |
| Full Occlusion | 48.3/59.5 | 53.1/66.0 | 34.8/51.6 | 37.5/46.7 | 47.1/59.7 | 45.0/54.0 | 49.7/59.9 | 56.2/68.5 | 51.4/62.4 | 54.0/65.8 | 56.1/68.2 | **58.0/70.2** |
| Low Illumination | 51.2/60.0 | 54.3/63.8 | 50.9/65.4 | 44.3/52.9 | 50.8/61.9 | 54.5/63.1 | 58.1/67.6 | 59.3/69.0 | 57.4/66.5 | 61.0/71.5 | 60.2/69.7 | **64.2/76.1** |
| Out-of-View | 45.6/52.4 | 47.9/55.6 | 37.0/44.9 | 38.5/46.5 | 42.3/50.2 | 48.0/53.9 | 49.6/56.4 | 50.6/57.9 | 50.0/57.7 | 52.6/60.6 | 52.5/60.2 | **54.0/62.5** |
| Partial Occlusion | 69.9/78.9 | 71.7/81.2 | 56.6/66.9 | 59.6/67.2 | 68.0/78.8 | 71.1/79.9 | 72.3/81.5 | 73.2/82.4 | 72.6/82.1 | 72.9/82.3 | 74.4/83.7 | **74.6/84.1** |
| Viewpoint Change | 64.7/72.1 | 69.3/77.0 | 52.1/60.3 | 55.3/61.6 | 59.7/69.6 | 67.8/75.7 | 68.6/75.6 | 70.3/78.7 | 70.1/78.1 | 71.8/79.7 | 71.0/79.3 | **73.8/82.5** |
| Scale Variation | 67.7/76.0 | 69.3/78.8 | 50.8/57.3 | 57.9/65.8 | 66.7/76.9 | 68.6/76.6 | 70.0/78.2 | 71.4/81.0 | 70.5/80.0 | 71.8/81.4 | 73.2/82.3 | **74.1/84.0** |
| Background Clutter | 55.3/64.8 | 57.0/67.0 | 48.6/63.3 | 46.8/57.4 | 50.7/61.7 | 58.1/67.7 | 59.2/69.0 | 61.0/71.3 | 59.4/69.4 | 61.3/72.0 | 62.1/71.7 | **64.8/76.3** |
| Motion Blur | 56.5/65.2 | 56.6/66.0 | 43.8/57.0 | 51.6/62.7 | 55.6/66.2 | 55.4/64.5 | 61.1/72.6 | 58.6/68.9 | 59.5/70.7 | 58.7/69.6 | 62.0/73.5 | **62.7/74.3** |
| Aspect Ration Change | 63.8/73.9 | 66.3/76.7 | 48.1/58.1 | 52.6/61.5 | 61.3/74.4 | 66.8/76.5 | 67.9/77.1 | 67.9/77.0 | 68.2/77.8 | 68.4/77.5 | 69.8/79.0 | **70.2/79.6** |
| Fast Motion | 64.7/69.8 | 66.7/73.0 | 54.5/59.8 | 54.3/56.8 | 63.2/70.7 | 66.8/71.8 | 67.9/73.3 | 69.4/75.3 | 68.7/74.8 | 69.9/76.0 | 69.8/76.0 | **71.1/77.5** |
| No Motion | 72.1/80.9 | 75.9/83.4 | 64.6/74.1 | 64.1/71.0 | 71.9/83.3 | 76.6/84.2 | 77.3/85.3 | 76.8/84.2 | 76.6/84.7 | 77.3/85.4 | 77.0/85.1 | **78.2/86.4** |
| Illumination Variation | 65.0/69.8 | 67.2/73.3 | 55.3/61.8 | 57.6/60.1 | 63.8/71.0 | 67.9/73.9 | 70.4/77.5 | 70.1/76.4 | 69.9/76.9 | 71.0/77.5 | 71.7/79.4 | **72.8/80.0** |
| Over Exposure | 67.0/70.9 | 69.4/75.5 | 58.5/64.8 | 58.1/60.8 | 65.7/73.1 | 70.4/76.4 | 72.2/78.9 | 71.8/78.0 | 72.0/78.9 | 72.8/79.2 | 73.0/80.4 | **73.7/80.8** |
| Background Object Motion | 64.0/72.7 | 66.0/75.2 | 54.3/65.5 | 54.8/62.8 | 61.2/71.8 | 65.8/74.2 | 67.3/76.1 | 68.4/77.4 | 67.3/76.3 | 69.2/78.6 | 69.6/78.7 | **71.3/81.1** |

**Comparison with UnTrack.** Here, we compare our method with the recent state-of-the-art multi-modal tracker, UnTrack (Wu et al., 2024). UnTrack is a unified tracker of a single set of parameters for three auxiliary modalities (e.g., VisEvent, DepthTrack and LasHeR datasets), which integrates the LoRA-tuning and Prompt-tuning techniques. As shown in Tab. 11, our method surpasses UnTrack across all multi-modal datasets.

### A.3 MORE ABLATION STUDIES.

**Comparison with sensitivity-aware sparse tuning (SPT).** In the proposed method, we exploit the parameter sensitivity to regularize parameter updates, instead of selecting the most sensitive parameters for sparse fine-tuning, e.g., SPT (He et al., 2023). In this section, we conduct experiments to

Table 10: Attribute performance on the **LasHeR** test set. The right superscript $^o$ represents the pre-trained OSTrack,and $^\dagger$ denotes the pre-trained DropTrack. The metric is the $RSR/RPR$.

| | OSTrack | DropTrack | SPT | APFNet | ProTrack$^o$ | ViPT$^o$ | ViPT$^\dagger$ | SDSTrack$^o$ | SDSTrack$^\dagger$ | Ours$^o$ | Ours$^\dagger$ |
|---|---|---|---|---|---|---|---|---|---|---|---|
| Illumination Variation | 37.7/42.2 | 40.2/46.2 | 34.1/35.7 | 38.6/41.0 | 28.4/30.7 | 35.8/38.0 | 38.9/43.6 | 48.2/54.2 | 43.8/50.1 | 49.1/55.7 | **53.5/62.1** |
| Aspect Ration Change | 43.4/51.4 | 45.1/53.7 | 35.9/40.7 | 37.7/42.8 | 42.5/49.7 | 49.4/58.5 | 49.5/59.5 | 50.1/60.4 | 50.9/61.9 | 51.9/62.4 | **55.5/67.4** |
| Background Clutter | 43.1/52.8 | 44.2/54.2 | 38.6/46.6 | 39.4/47.5 | 41.4/50.1 | 52.0/64.5 | 50.7/63.6 | 51.3/63.6 | 55.0/69.6 | 55.7/69.1 | **57.9/72.3** |
| Camera Motion | 37.4/45.6 | 39.4/49.1 | 31.9/38.9 | 34.6/41.7 | 40.7/50.7 | 46.6/58.7 | 47.0/58.7 | 46.5/59.2 | 47.5/60.6 | 48.0/60.4 | **51.7/65.3** |
| Deformation | 47.8/57.6 | 47.5/57.6 | 45.5/54.1 | 48.3/56.9 | 46.7/54.8 | 55.7/66.7 | 55.6/67.5 | 56.3/68.8 | 59.5/73.4 | 60.3/73.2 | **61.7/75.2** |
| Frame Lost | 47.3/53.1 | 60.5/68.4 | 45.0/45.1 | 44.0/45.3 | 64.6/74.8 | 64.8/75.4 | 66.6/76.5 | 60.6/69.5 | **66.4/77.0** | 62.8/71.5 | 62.6/72.3 |
| Fast Motion | 45.2/54.9 | 46.5/56.6 | 38.7/45.6 | 40.1/47.6 | 44.4/53.2 | 51.4/62.4 | 51.6/63.2 | 52.8/64.9 | 53.7/66.7 | 56.0/68.7 | **58.0/71.6** |
| High Illumination | 41.4/51.1 | 45.5/58.2 | 46.5/58.8 | 50.4/63.1 | 46.6/58.5 | 54.6/68.2 | 57.5/73.1 | 55.1/69.1 | 58.8/75.2 | 56.8/70.7 | **64.5/81.3** |
| Hyaline Occlusion | 41.8/43.5 | 43.7/47.5 | 38.4/36.9 | 40.7/41.7 | 31.7/30.6 | 42.1/44.8 | 44.6/47.4 | 52.7/58.8 | 49.8/56.1 | 49.8/55.6 | **53.5/61.2** |
| Low Illumination | 34.5/41.7 | 37.1/45.0 | 33.5/39.6 | 36.7/43.8 | 34.0/39.8 | 41.4/49.3 | 43.6/53.3 | 43.8/53.4 | 47.2/58.0 | 48.9/60.1 | **52.5/65.2** |
| Low Resolution | 34.9/47.4 | 38.5/51.8 | 28.3/39.9 | 30.4/41.5 | 34.2/45.7 | 41.8/56.4 | 42.9/58.0 | 42.5/57.2 | 45.0/60.9 | 47.2/62.8 | **48.4/64.8** |
| Motion Blur | 40.3/49.5 | 41.6/51.4 | 33.8/41.4 | 35.8/44.0 | 38.5/47.1 | 46.1/57.0 | 45.9/57.3 | 46.2/57.2 | 48.7/61.2 | 50.3/62.2 | **53.3/66.3** |
| No Occlusion | 62.8/76.7 | 64.7/79.9 | 59.2/74.4 | 63.0/77.9 | 62.4/78.3 | 68.3/83.3 | 68.0/83.6 | 70.8/86.9 | 70.7/87.3 | **74.0/90.2** | 72.9/88.8 |
| Out-of-View | 38.9/47.7 | 40.8/51.2 | 34.4/42.1 | 35.6/43.2 | 39.1/47.7 | 45.7/56.8 | 46.5/58.4 | 47.5/59.6 | 49.3/62.3 | 49.5/62.0 | **53.7/67.7** |
| Partial Occlusion | 43.6/53.6 | 45.1/55.7 | 36.5/43.8 | 38.7/46.8 | 43.5/52.7 | 50.5/62.1 | 50.3/62.5 | 50.6/62.9 | 52.7/66.1 | 54.1/67.0 | **56.8/70.8** |
| Similar Appearance | 40.8/49.5 | 40.8/50.0 | 34.1/40.5 | 35.2/41.8 | 40.2/48.0 | 46.6/57.2 | 45.0/55.6 | 46.3/57.1 | 48.2/60.1 | 49.6/61.3 | **52.5/64.7** |
| Scale Variation | 46.4/56.8 | 48.0/59.1 | 39.3/47.5 | 41.8/50.6 | 46.4/56.6 | 52.4/64.4 | 52.7/65.1 | 53.0/65.8 | 54.7/68.4 | 56.5/69.7 | **58.7/72.8** |
| Thermal Crossover | 43.7/53.8 | 44.9/55.8 | 37.3/45.5 | 39.4/48.1 | 43.1/52.9 | 49.9/61.5 | 50.7/63.5 | 50.7/63.2 | 52.8/66.2 | 53.9/66.6 | **57.6/72.0** |
| Total Occlusion | 41.7/51.9 | 41.6/51.8 | 35.2/42.5 | 37.4/45.5 | 38.7/47.2 | 46.4/57.5 | 47.2/59.3 | 47.8/60.0 | 48.6/60.9 | 51.8/64.4 | **52.9/66.3** |

Table 11: Comparison of the tracking performance between UnTrack and our method based on VisEvent, DepthTrack and LasHeR datasets.

| Exp. | Base Model | VisEvent | | | | DepthTrack | | | LasHeR | | | |
|---|---|---|---|---|---|---|---|---|---|---|---|---|
| | | RSR | OP$_{0.5}$ | OP$_{0.75}$ | RPR | Pr | Re | F-score | RSR | OP$_{0.5}$ | OP$_{0.75}$ | RPR |
| UnTrack | OSTrack | 59.1 | 69.7 | 57.1 | 73.2 | 61.1 | 60.8 | 61.0 | 52.5 | 63.7 | 42.9 | 65.3 |
| Ours | OSTrack | 62.0 | 73.7 | 60.9 | 76.5 | 66.1 | 65.0 | 65.5 | 56.3 | 68.0 | 48.5 | 69.5 |
| **Improvement** | OSTrack | **+2.9** | **+4.0** | **+3.8** | **+3.3** | **+5.0** | **+4.2** | **+4.5** | **+3.8** | **+4.3** | **+5.6** | **+4.2** |
| UnTrack | DropTrack | 62.0 | 73.9 | 59.8 | 76.7 | 63.7 | 63.7 | 63.7 | 53.7 | 65.0 | 43.6 | 67.0 |
| Ours | DropTrack | 63.2 | 75.4 | 61.4 | 78.1 | 67.8 | 66.3 | 67.0 | 58.8 | 71.5 | 50.2 | 73.0 |
| **Improvement** | DropTrack | **+1.1** | **+1.5** | **+1.6** | **+1.4** | **+4.1** | **+2.6** | **+3.3** | **+5.1** | **+6.5** | **+6.6** | **+6.0** |

investigate our performance against SPT. Note that the training configuration of those methods is exactly the same, except for the trainable parameter ratio. As shown in Tab. 12, the experimental results show that applying sparse fine-tuning to the cross-modal adaptation cannot achieve considerable performance.

Table 12: Comparison of the tracking performance between SPT and our regularized fine-tuning method based on the DepthTrack dataset, using the pre-trained OSTrack as the base model. We have set up a series of trainable parameter ratios $\tau$ of SPT (top-$\tau$ sensitive parameters) to fully explore its adaption effect.

| Exp. | Pr | Re | F-score |
|---|---|---|---|
| w/o fine-tuning ($\tau = 0\%$) | 38.2 | 36.0 | 37.1 |
| full fine-tuning ($\tau = 100\%$) | 61.7 | 61.5 | 61.6 |
| SPT ($\tau = 50\%$) | 61.1 | 60.9 | 61.0 |
| SPT ($\tau = 20\%$) | 60.7 | 59.7 | 60.2 |
| SPT ($\tau = 10\%$) | 56.2 | 52.8 | 54.5 |
| Ours ($\tau = 100\%$) | 66.1 | 65.0 | 65.5 |

**Comparison with low rank adaptation tuning (LoRA).** Low-rank adaptation tuning (e.g., LoRA (Hu et al., 2021)) is a widely used parameter-efficient fine-tuning method. Hence, we also conduct experiments to investigate its cross-modal transfer performance. As summarized in Tab. 13, the results indicate that applying LoRA to cross-modal tracking yields limited performance gains. We attribute this to the strong constraints imposed on pre-trained weights, similar to other parameter-efficient methods like prompt tuning and visual adapters, which hinder effective transfer learning.

**Smaller learning rates.** In the proposed method, we exploit the sensitivity-aware momentum gradients to bolster the stability and coherence of multi-modal representations. A straightforward smoothing approach might involve using smaller learning rates. In this section, we explore the impact of smaller learning rates on performance. As shown in Tab. 14, the results indicate that a smaller learning rate (i.e., $lr = 10^{-5}$) offers negligible improvement. Further reduction (i.e., $lr = 10^{-6}$) leads to performance degradation. These findings suggest that smaller learning rates fail to effectively address the overfitting issue in cross-modal adaptation,

Table 13: Comparison of the tracking performance between LoRA and our method based on FE108, DepthTrack and LasHeR datasets, using the pre-trained OSTrack. And we have set up a series of ranks ($r$) of LoRA to fully explore its adaptability.

| Exp. | FE108 | | | | DepthTrack | | | LasHeR | | | |
|---|---|---|---|---|---|---|---|---|---|---|---|
| | RSR | $OP_{0.5}$ | $OP_{0.75}$ | RPR | Pr | Re | F-score | RSR | $OP_{0.5}$ | $OP_{0.75}$ | RPR |
| $r$=2 | 64.1 | 82.7 | 41.0 | 91.1 | 61.7 | 61.0 | 61.3 | 54.1 | 64.5 | 43.6 | 64.9 |
| $r$=4 | 64.7 | 83.7 | 42.4 | 91.4 | 61.8 | 61.1 | 61.4 | 54.3 | 64.6 | 43.7 | 65.2 |
| $r$=8 | 64.5 | 83.8 | 41.1 | 91.3 | 61.6 | 61.0 | 61.2 | 54.3 | 64.6 | 43.8 | 65.1 |
| Ours | 67.5 | 87.0 | 44.4 | 95.5 | 66.1 | 65.0 | 65.5 | 56.3 | 68.0 | 48.5 | 69.5 |

**as they treat all parameters uniformly without focusing on the update of sensitive/high-risk parameters.**

Table 14: We explored the effect of smaller learning rates $lr$. All experiments were conducted on the LasHeR dataset, using the pre-trained OSTrack as the base model.

| Exp. | RSR | $OP_{0.5}$ | $OP_{0.75}$ | RPR |
|---|---|---|---|---|
| $lr = 10^{-4}$ | 54.3 | 64.1 | 45.3 | 65.4 |
| $lr = 10^{-5}$ | 53.7 | 64.8 | 45.0 | 65.9 |
| $lr = 10^{-6}$ | 52.5 | 63.7 | 44.2 | 65.1 |
| Ours | 56.3 | 68.0 | 48.5 | 69.5 |

**Effectiveness across different resolutions.** **As only OSTrack-256 and DropTrack-384 model weights are officially available, other resolution settings were not discussed in the previous manuscript. In transformer-based trackers, resolution profoundly affects position embeddings and tokens, both essential for object-aware relation encoding. To verify whether the proposed method works effectively across resolutions, we conducted the experiments in Tab. 15, testing the OSTrack at 384 resolution and the DropTrack at 256 resolution. The results confirm that resolution changes substantially impact tracking performance, aligning with our claim.**

Table 15: Ablation of the cross-resolution performance of our method based on the VisEvent, Depth-Track and LasHeR datasets.

| Exp. | VisEvent | | | | DepthTrack | | | LasHeR | | | |
|---|---|---|---|---|---|---|---|---|---|---|---|
| | RSR | $OP_{0.5}$ | $OP_{0.75}$ | RPR | Pr | Re | F-score | RSR | $OP_{0.5}$ | $OP_{0.75}$ | RPR |
| OSTrack-256 (Official) | 62.0 | 73.7 | 60.9 | 76.5 | 66.1 | 65.0 | 65.5 | 56.3 | 68.0 | 48.5 | 69.5 |
| OSTrack-384 | 60.7 | 72.1 | 59.5 | 74.8 | 65.2 | 64.7 | 64.9 | 55.3 | 66.7 | 47.7 | 68.4 |
| DropTrack-384 (Official) | 63.2 | 75.4 | 61.4 | 78.1 | 67.8 | 66.3 | 67.0 | 58.8 | 71.5 | 50.2 | 73.0 |
| DropTrack-256 | 61.7 | 73.3 | 60.2 | 76.4 | 66.3 | 64.6 | 65.4 | 57.5 | 69.8 | 49.0 | 71.4 |

## A.4 VISUALIZATION OF TRACKING RESULTS.

In Fig 11, we present a more qualitative comparison of tracking results between our method and existing approaches. We can observe that the proposed method shows better regression capability in challenging conditions.

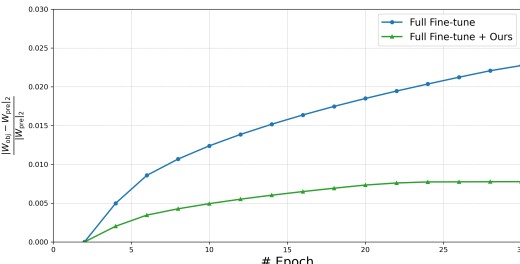

Figure 10: **Relation $L_2$ distance between our fine-tuning method and the full fine-tuning on the DepthTrack and pre-trained OSTrack model in the parameter space. Our method significantly reduces weight deviation, indicating improved retention of the pretrained knowledge while achieving the desired adaptation.**

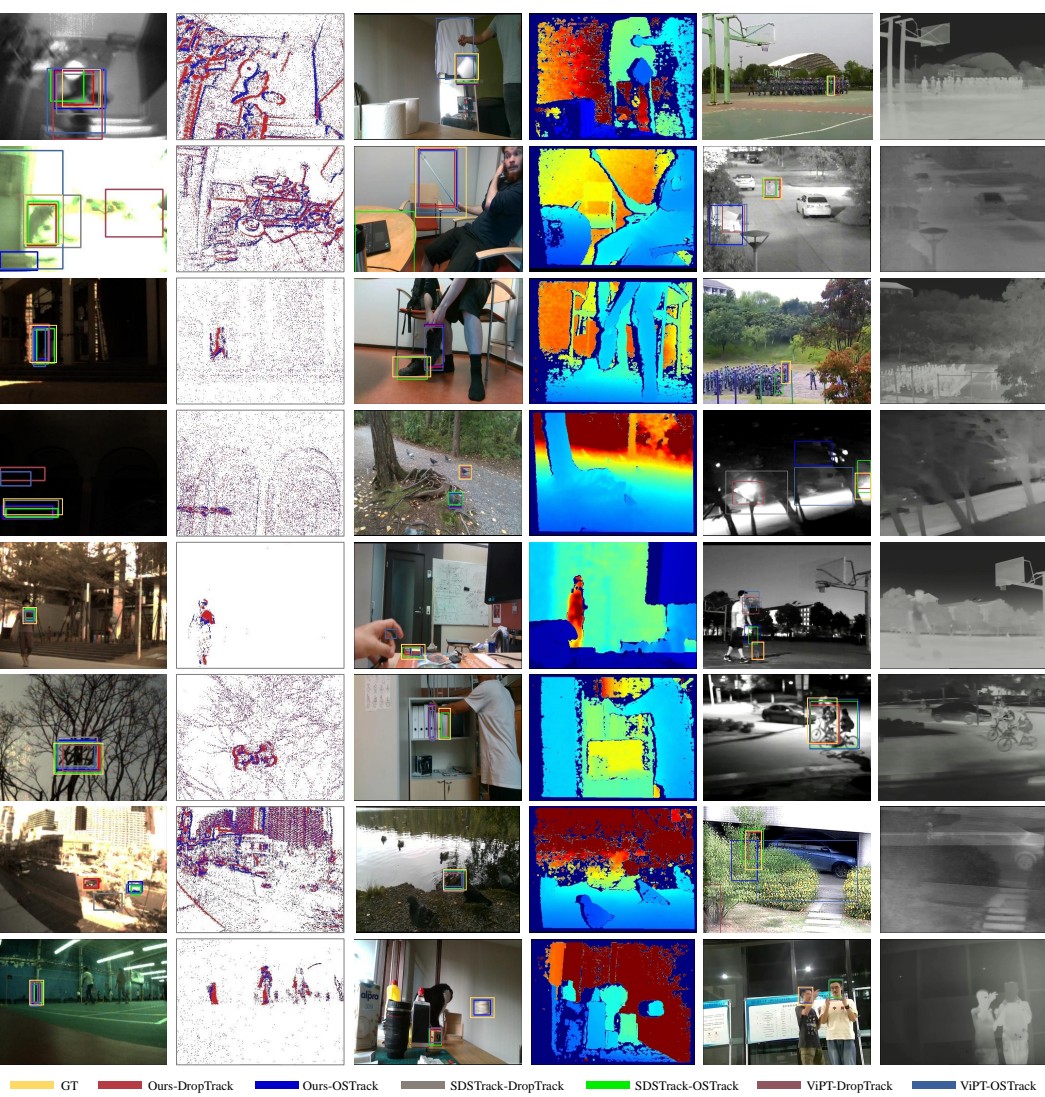

Figure 11: Visual comparisons of the tracking performance of different methods on the (**Left**) RGB-Event, (**Middle**) RGB-Depth and (**Right**) RGB-Thermal datasets.

