# OpenReview forum: "Learning Effective Multi-modal Trackers via Modality-Sensitive Tuning"
_ICLR.cc/2025/Conference — ICLR 2025 Conference Withdrawn Submission_

### Official Review · Reviewer_Dsbe · 2024-10-20

**Soundness:** 2
**Presentation:** 3
**Contribution:** 2
**Rating:** 5
**Confidence:** 5

**Summary:**

This paper tries to propose a modality-sensitive tuning technique for adapting the RGB pre-trained models to the downstream tasks (multi-modal tracking in this paper). From my perspective, this is an interesting motivation since the current community of multi-modal tracking is focusing on design unifed models. Besides, I believe this paper is well-written and various experimental results are provided to demonstrate its effectiveness.

**Strengths:**

1. This paper is well-written and easy to follow.
2. The motivation is reasonable and also highly related to recent trend of the multi-modal tracking community.
3. Various experiments present that the proposed technique is working with better performance witnessed on several benchmarks.

**Weaknesses:**

1.	Where are the results reported in Figure 1 from, LasHeR, DepthTrack, VisEvent or other datasets? In this figure, SDSTrack significantly performs worse than ViPT which against my intuition and the official paper as well. Displaying the results on larger dataset like VisEvent or DepthTrack or LasHeR should be better.
2.	A mistake in Table 3, the title ‘SR” is missed.
3.	In Table 5, when fully finetuning ViPT, the results degrade. But it grows as reported in the official manuscript.
4.	As claimed, the current tuning techniques are over- or under-fitting. But it’s not demonstrated that with the proposed technique, the methods are not over- or under-fitting. If the authors want to clarify this point through figure 1, I will suggest the authors to add this relation in the paper. Additionally, ViPT is officially trained 60 epochs and I would like to see the curves at 60 or even more epochs.
5.	The motivation is smoothing the adaption. For this purpose, the most straightforward way is utilizing smaller learning rates. But it’s not investigated in the current version.
6.	As to the definition of modality sensitivity, the training objective is employed as a criterion, which is more like measuring the model-sensitivity rather than the multi-modal sensitivity.
7.	In generally, from my perspective, the proposed method seems an adaptive Exponential Moving Average (EMA), where this adaptation is achieved by measuring the model sensitivity. Thus, it has limited relation with multi-modal tracking and does not provide any insight for this task.

**Questions:**

1.	Where are the results reported in Figure 1 from, LasHeR, DepthTrack, VisEvent or other datasets? In this figure, SDSTrack significantly performs worse than ViPT which against my intuition and the official paper as well. Displaying the results on larger dataset like VisEvent or DepthTrack or LasHeR should be better.
2.	In Table 5, when fully finetuning ViPT, the results degrade. But it grows as reported in the official manuscript.
3.	As claimed, the current tuning techniques are over- or under-fitting. But it’s not demonstrated that with the proposed technique, the methods are not over- or under-fitting. If the authors want to clarify this point through figure 1, I will suggest the authors to add this relation in the paper. Additionally, ViPT is officially trained 60 epochs and I would like to see the curves at 60 or even more epochs.
4.	The motivation is smoothing the adaption. For this purpose, the most straightforward way is utilizing smaller learning rates. But it’s not investigated in the current version.
5.	As to the definition of modality sensitivity, the training objective is employed as a criterion, which is more like measuring the model-sensitivity rather than the multi-modal sensitivity.
6.	In generally, from my perspective, the proposed method seems an adaptive Exponential Moving Average (EMA), where this adaptation is achieved by measuring the model sensitivity. Thus, it has limited relation with multi-modal tracking and does not provide any insight for this task.

---

> ### Author Response · Authors · 2024-11-24
>
> **Q1:** Where are the results reported in Figure 1 from, LasHeR, DepthTrack, VisEvent or other datasets? In this figure, SDSTrack significantly performs worse than ViPT which against my intuition and the official paper as well. Displaying the results on larger dataset like VisEvent or DepthTrack or LasHeR should be better. As claimed, the current tuning techniques are over- or under-fitting. But it’s not demonstrated that with the proposed technique, the methods are not over- or under-fitting. If the authors want to clarify this point through figure 1, I will suggest the authors to add this relation in the paper. Additionally, ViPT is officially trained 60 epochs and I would like to see the curves at 60 or even more epochs.
>
> **A:** We appreciate the reviewer for highlighting this deficiency. In response, **we have revised Figure.1** using LasHeR dataset, showcasing extended training phases with supplemented training performance as a reference. As illustrated in the revised Figure.1, full fine-tuning exhibits severe over-fitting (evidenced by a significant gap between training and testing performance), while parameter efficient fine-tuning demonstrates under-fitting (reflecting its limited capacity to model multi-modal data during training). **We believe the revised figure effectively showcases the limitations of existing methods and clearly underscores the contributions of our approach.**
>
> **Q2:** In Table 5, when fully finetuning ViPT, the results degrade. But it grows as reported in the official manuscript.
>
> **A:** Thanks for your question. In the the official manuscript (**Table.4, FFT setting**), the results clearly indicate that the full fine-tuning degrades the performance, which is completely consistent with our findings (**Table.5, ViPT+F-Tune. setting**).
>
> **Q3:** The motivation is smoothing the adaption. For this purpose, the most straightforward way is utilizing smaller learning rates. But it’s not investigated in the current version.
>
> **A:** A straightforward smoothing approach might involve using smaller learning rates. In response, we investigate the impact of smaller learning rates on performance, as shown in **Table.14 (in Appendix A.3)**. The results suggest that smaller learning rates are insufficient to address the over-fitting issue in cross-modal adaptation, as they treat all parameters uniformly without focusing on the update of sensitive/high-risk parameters.
>
> **Q4:** As to the definition of modality sensitivity, the training objective is employed as a criterion, which is more like measuring the model-sensitivity rather than the multi-modal sensitivity.
>
> **A:** Thanks for your question. In our framework, parameter sensitivity quantifies each parameter's contribution to the reduction of the training objective. To preserve the generalization of pre-trained RGB knowledge during cross-modal transfer, we suppress updates to parameters highly sensitive to specific auxiliary modalities (e.g., event, depth, thermal). Due to the distinct data distributions of these modalities, parameter sensitivity varies significantly during training. To provide greater clarity, we have supplemented a modality-aware parameter-wise sensitivity diagram in the **updated Figure.3.** **This diagram reveals that parameters in the auxiliary branch exhibit strong similarities within the same modality and clear differences across modalities, highlighting the modality-driven parameter preferences inherent in cross-modal fine-tuning.**
>
> **Q5:** In generally, from my perspective, the proposed method seems an adaptive Exponential Moving Average (EMA), where this adaptation is achieved by measuring the model sensitivity. Thus, it has limited relation with multi-modal tracking and does not provide any insight for this task.
>
> **A:**  We thank the reviewer for the valuable comment. To the best of our knowledge, this study comprehensively revisits and highlights the ill-fitting issues encountered in cross-modal tracking when adapting foundational models. It introduces a novel self-regularizing cross-modal transfer framework that significantly enhances generalization and operates independently of existing full fine-tuning and parameter-efficient fine-tuning methods. Our approach is intuitive, consistent, and efficient, optimizing the learning process from both inter-structural and inter-temporal perspectives. **Importantly, we must emphasize that modality sensitivity more accurately captures the core essence of sensitivity compared to model- or data-based sensitivity.** This is because the modality-driven distribution shift plays a pivotal role in the training process, making it directly relevant to the unique challenges of cross-modal tracking. Therefore, our work not only provides valuable insights into this task but also effectively tackles its core limitations.

---

> ### Comment · Reviewer_Dsbe · 2024-11-24
> **Learning Effective Multi-modal Trackers via Modality-Sensitive Tuning**
>
> Thank you for your rebuttal.
> I have read all the reviews and responses. I think the authors have carefully sloved most the questions.
> First of all, I apologise for my mistake in question 2.
> However, I think the authors do not solve my main question in Q5, as claimed ' Importantly, we must emphasize that modality sensitivity more accurately captures the core essence of sensitivity compared to model- or data-based sensitivity', this means authors themselves are still not sure about the sensitivity.
> This is the main contribution, but unfortuantely,  I'm not convinced and keep on my original opinion.
> But according to the rebuttal, I'm considering improving the score.

---

> > ### Author Response · Authors · 2024-11-25
> >
> > We thank the reviewer Dsbe for the opportune and insightful comments.
> > As stated, the assertion that 'this means authors themselves are still not sure about the sensitivity' represents a misunderstanding of our position. **We firmly affirm the motivation and definition of modality sensitivity**.
> > - **Explicit modality sensitivity:** In this paper, we delineate the sensitivity as the contributions/biases of parameters within the same pre-trained model toward data-derived learning objectives. During the transfer learning process, this sensitivity manifests as a clear modality-awareness. As illustrated **in the updated Figure 3**, it is evident that parameters in the auxiliary branch show strong similarities within the same modality and distinct divergences across different modalities. This **highlights the inherent modality-driven parameter preferences** crucial for effective cross-modal fine-tuning. Furthermore, we analyze parameter-wise sensitivity across different model locations, as shown **in Figure.7**. Specifically, the clustering of sensitive parameters in certain areas **reflects an over-reliance on specific parameters.** After applying sensitivity penalties, the patterns become more balanced, indicating that **reducing location bias enhances the model’s generalization.**

---

> > > ### Comment · Reviewer_Dsbe · 2024-11-26
> > > **Learning Effective Multi-modal Trackers via Modality-Sensitive Tuning**
> > >
> > > Thank you for your answer.
> > > Figure 3 is clear to show the differences between multiple modalties.
> > > But, what you have done is computing the loss and adaptively generating the gradients, this is definiately the model sensitivity rather than the modal sensitivity.
> > > But generally, I think you deserve higher score, so i change to 5

---

> ### Author Response · Authors · 2024-11-26
>
> We are grateful to Reviewer Dsbe for the opportune comments and the improved evaluation. As claimed, “But, what you have done is computing the loss and adaptively generating the gradients, this is definiately the model sensitivity rather than the modal sensitivity.” To this end, we provide a more nuanced explanation of modality sensitivity. The sensitivity in this context refers to the responsiveness of **a model or its parameters (A)** to **a specific object (B)** during the transfer/fine-tuning process. Inherently, **sensitivity itself involves the model**, an aspect **can be omitted** in the item (i.e., model sensitivity). Moving beyond this, our method is standardized, where each parameter is initialized with the same pre-trained model **(which ensures model consistency across experiments)**. Notably, **the primary focus of sensitivity** is directed towards **the specific objects (B)**, such as modality, instance or other influential factors. In this work, our method adaptively captures and harnesses specific modality sensitivities within the data to refine the cross-modal update process, which we term **modality sensitivity** (in Figure.3 and Figure.7). This distinct approach underscores our contribution to understanding and optimizing interaction across modalities.

---

> ### Author Response · Authors · 2024-11-30
>
> Hi **Reviewer Dsbe**,
>
> In our responses, we have further clarified the modality sensitivity. Moreover, we have made every effort to address your concerns and would greatly appreciate any further feedback. Given that there are **only a few days remaining for the discussion**, we would be truly grateful if you could take a few minutes to review our response. Thank you for your time and consideration!

---

> ### Author Response · Authors · 2024-12-02
>
> Hi **Reviewer Dsbe**,
>
> In our responses, we have gone to great lengths to clarify the modality sensitivity and would greatly appreciate any further feedback. Given that the discussion period is running out, we respectfully request that you take a few moments to review our detailed response. We firmly believe that our clarifications and efforts warrant a reassessment, and we sincerely hope this will result in a more favorable evaluation of our work.

---

> ### Author Response · Authors · 2024-12-03
>
> Hi **Reviewer Dsbe**,
>
>
> We have diligently clarified the issues regarding modality sensitivity and welcome any additional feedback. As the discussion period is quickly concluding, we respectfully request that you take a few moments to review our detailed response. We are confident that our thorough clarifications and dedicated efforts justify a reassessment. We earnestly anticipate that this will lead to a more favorable evaluation of our work.

---

### Official Review · Reviewer_x1zj · 2024-10-24

**Soundness:** 2
**Presentation:** 2
**Contribution:** 2
**Rating:** 6
**Confidence:** 5

**Summary:**

This paper presents a modality sensitivity-aware tuning framework (MST) that improves the fine-tuning process to enhance tracking performance. It primarily introduces the concept of parameter modality sensitivity and utilizes it to standardize parameter updates, serving as a measure for multimodal adaptation. The proposed approach achieves commendable results across multiple multimodal tracking tasks and datasets.

**Strengths:**

1. A new modality sensitivity-aware framework, MST, is proposed, optimizing the learning dynamics of cross-modal trackers from two key perspectives: modeling parameter modality sensitivity and performing adaptive tuning that is sensitive to modality, introducing a novel fine-tuning method.
2. The use of parameter modality sensitivity to standardize parameter updates is proposed.
3. A self-integrating weight strategy is introduced to enhance the stability and consistency of multimodal representations, contributing to improved model generalization capabilities.

**Weaknesses:**

1. The approach mentioned in the abstract of using RGB pre-training to adapt to multimodal tasks is a common practice in the field and does not present any challenges. It fails to address the difficulties and new issues encountered in this task.
2. Some sections are overly complex, and the modeling of modality sensitivity and the derivation of formulas lack readability and comprehensibility.
3. This "parameter tuning" method does not differ fundamentally from approaches like ViPT and SDSTrack, which require task-specific training and fine-tuning. Unlike OneTracker, which can adapt to all tasks with a single fine-tuning, this method does not demonstrate significant advancements or improvements.
4. What exactly is modality sensitivity modeling in multimodal tracking? What does modality sensitivity-aware tuning for multimodal trackers entail? Although the paper discusses gradient computation and parameter-related formulas, it fails to clarify these concepts, and there is no reference to related research or underlying principles.
5. Issues in the experimental section: a) The evaluation of tracker speed lacks information on the evaluation platform used. b) For the DepthTrack dataset, the evaluation metrics of precision-recall (pr) and success rate (sr) are unclear, along with the significance of OP0.5 and OP0.75. Moreover, the reported pr values do not match those in the official tracker papers, which state the evaluation metrics as F-score, Recall, and Precision. c) The paper mentions that SDSTrack and ViPT use OSTrack as a pre-training result, but it is known that the official evaluations also use OSTrack for pre-training. The results reported in the paper do not align with the official findings.

**Questions:**

1. In the section "Computational Cost and Inference Speed," why are the parameters of the trackers obtained from OSTrack and DropTrack identical despite using different pre-training methods?
2. What is Modality Sensitivity, modality sensitivity modeling and modality sensitivity-aware tuning? The definition is vague and lacks explanatory depth.
3. Why does DepthTrack use precision-recall (pr) and success rate (sr) as evaluation metrics, and what do OP0.5 and OP0.75 signify?

---

> ### Author Response · Authors · 2024-11-24
>
> **Q1:** The approach mentioned in the abstract of using RGB pre-training to adapt to multimodal tasks is a common practice in the field and does not present any challenges. It fails to address the difficulties and new issues encountered in this task.
>
> **A:** We kindly **disagree** with this assertion. A fundamental limitation lies in its susceptibility to over-fitting, stemming from the mismatch between the scarcity of large-scale auxiliary datasets and the substantial demands of cross-modal transfer. Our work effectively mitigates the over-fitting, enhancing the adaptability of foundational models for cross-modal tracking.
>
> **Q2:** (i) Some sections are overly complex, and the modeling of modality sensitivity and the derivation of formulas lack readability and comprehensibility. What does modality sensitivity-aware tuning for multimodal trackers entail? (ii) Although the paper discusses gradient computation and parameter-related formulas, it fails to clarify these concepts, and there is no reference to related research or underlying principles.
>
> **A:** .
> - **For question (i) :** We discuss the modeling of modality sensitivity from two perspectives:
>   - **Explicit modality sensitivity:** In this paper, we delineate the sensitivity as the contributions/biases of parameters within the same pre-trained model toward data-derived learning objectives. During the transfer learning process, this sensitivity manifests as a clear modality-awareness. As illustrated **in the updated Figure 3**, it is evident that parameters in the auxiliary branch show strong similarities within the same modality and distinct divergences across different modalities. This **highlights the inherent modality-driven parameter preferences** crucial for effective cross-modal fine-tuning.
>   - **Explicit location sensitivity:** We analyze parameter-wise sensitivity across different model locations, as shown **in Figure.7**. Specifically, the clustering of sensitive parameters in certain areas **reflects an over-reliance on specific parameters.** After applying sensitivity penalties, the patterns become more balanced, indicating that **reducing location bias enhances the model’s generalization**.
> - **For question (ii) :** In the **Remark** (Section 3.3), we have included relevant references and explicitly clarified how our work builds upon and relates to prior research. Furthermore, we conducted additional experiments of alternative tuning methods in Appendix A.3.
>
> **Q3:** (i) This "parameter tuning" method does not differ fundamentally from approaches like ViPT and SDSTrack, which require task-specific training and fine-tuning. (ii) Unlike OneTracker, which can adapt to all tasks with a single fine-tuning, this method does not demonstrate significant advancements or improvements.
>
> **A:** .
> - **For question (i) :** We kindly **disagree** with this comment. The essence of parameter-efficient fine-tuning lies in keeping pre-trained parameters fixed while introducing additional tunable parameters for downstream tasks. To this end, the rigid constraints imposed on pre-trained weights result in sub-optimal transfer learning, **as clearly illustrated in the revised Figure.1.**
> - **For question (ii) :** As the code for OneTracker are currently unavailable, we refer to a recently similar work, **UnTrack [R1]**. UnTrack is a unified tracker with a single set of parameters for three auxiliary modalities. As shown in Table.C1, our method significantly outperforms UnTrack.
>
> #### **Table C1:** Comparison with UnTrack
> | Exp. | Base Model ||| VisEvent ||| DepthTrack ||| LasHeR |||
> | :---: | :---: | :---: | :---: | :---: | :---: | :---: | :---: | :---: | :---: | :---: | :---: | :---: |
> |  |  | RSR | OP$_{0.5}$ | OP$_{0.75}$ | RPR | Pr | Re | F-score | RSR | OP$_{0.5}$ | OP$_{0.75}$ | RPR |
> | UnTrack | OSTrack | 59.1 | 69.7 | 57.1 | 73.2 | 61.1 | 60.8 | 61.0 | 52.5 | 63.7 | 42.9 | 65.3 |
> | Ours | OSTrack | 62.0 | 73.7 | 60.9 | 76.5 | 66.1 | 65.0 | 65.5 | 56.3 | 68.0 | 48.5 | 69.5 |
> | Improvement | OSTrack | **+2.9** | **+4.0** | **+3.8** | **+3.3** | **+5.0** | **+4.2** | **+4.5** | **+3.8** | **+4.3** | **+5.6** | **+4.2** |
> | UnTrack | DropTrack | 62.0 | 73.9 | 59.8 | 76.7 | 63.7 | 63.7 | 63.7 | 53.7 | 65.0 | 43.6 | 67.0 |
> | Ours | DropTrack | 63.2 | 75.4 | 61.4 | 78.1 | 67.8 | 66.3 | 67.0 | 58.8 | 71.5 | 50.2 | 73.0 |
> | Improvement | DropTrack | **+1.1** | **+1.5** | **+1.6** | **+1.4** | **+4.1** | **+2.6** | **+3.3** | **+5.1** | **+6.5** | **+6.6** | **+6.0** |
>
> [R1] Single-model and any-modality for video object tracking, CVPR, 2024

---

> > ### Author Response · Authors · 2024-11-24
> >
> > **Q4:** Issues in the experimental section: (i) The evaluation of tracker speed lacks information on the evaluation platform used. (ii) For the DepthTrack dataset, the evaluation metrics of precision-recall (pr) and success rate (sr) are unclear, along with the significance of OP0.5 and OP0.75. Moreover, the reported pr values do not match those in the official tracker papers, which state the evaluation metrics as F-score, Recall, and Precision.(iii) The paper mentions that SDSTrack and ViPT use OSTrack as a pre-training result, but it is known that the official evaluations also use OSTrack for pre-training.
> >
> > **A:**  We thank the reviewer for the insightful comments.
> > - **For question (i) :** We have supplemented the platform information.
> > - **For question (ii) :** On the DepthTrack, we have adopted **F-score, Recall, and Precision** as the evaluation metrics. Additionally, we must clarify that the evaluation metrics presented in the initial submission, i.e., representative success rate (RSR), representative precision rate (RPR), and overlap precision ($OP_T$) are widely recognized and commonly used in the field of object tracking [R1~R3]. Specifically, success rate cares the frame of that overlap between ground truth and predicted bounding box is larger than a threshold; We employ the area under curve (AUC) of a success rate plot as representative success rate (RSR). Precision rate focuses on the frame of that the center distance between ground truth and predicted bounding box within a given threshold; We use the precision rate score associated with a 20-pixel threshold as representative precision rate (RPR). $OP_T$ represents success rate with $T$ as the threshold, $OP_{0.5}$\, and $OP_{0.75}$ represent the success rates under moderate and challenging conditions, respectively.
> >
> > - **For question (iii) :** I can **confirm unequivocally** that SDSTrack and ViPT do not involve training foundational models on RGB tracking data. Instead, they leverage a pre-trained OSTrack model, focusing on parameter-efficient fine-tuning. Notably, the OSTrack model has already undergone extensive training to convergence on RGB-based tracking datasets such as COCO, LaSOT, GOT-10k, and TrackingNet.
> >
> > **Q6:** In the section "Computational Cost and Inference Speed," why are the parameters of the trackers obtained from OSTrack and DropTrack identical despite using different pre-training methods?
> >
> > **A:** As detailed in the initial submission (**experimental settings**), both OSTrack and DropTrack utilize ViT-B/16 as their backbone, which constitutes the majority of their parameters. The primary difference between these models lies in the scale of datasets used during pre-training.

---

> > ### Author Response · Authors · 2024-12-03
> >
> > Hello **Reviewer x1zj**,
> >
> > We have thoroughly addressed your concerns in our revised manuscript and have supplemented our submission with additional experiments to underscore the efficacy of our method. We are troubled by what seems to be a lack of engagement on your part, reflected both in the undeservedly low score and the absence of interaction during the rebuttal phase. We strongly urge you to reevaluate the final score, considering the extensive revisions and data we have provided in our response.

---

> ### Author Response · Authors · 2024-11-25
> **Looking forward to discussion**
>
> Dear Reviewer x1zj,
>
> We sincerely thank you for devoting time to this review and providing valuable comments. Based on the comments, we have revised our manuscript to include the following changes:
> - We have further clarified the motivation and definition of modality sensitivity.
> - We have reproduced UnTrack to enable a more comprehensive comparison.
> - We have adapted F-score, Recall, and Precision as the evaluation metrics on the DepthTrack datatset.
> - We have clarified the issues and misunderstandings in the experimental section.
> We will actively participate in the Author-Reviewer discussion session. Please don't hesitate to let us know of any additional comments on the manuscript or the changes.
>
> Best regards,
>
> The Authors

---

> ### Author Response · Authors · 2024-11-28
>
> Hi **Reviewer x1zj**,
>
> In previous responses and the revised manuscript, we have further clarified the modality sensitivity and included additional experiments to demonstrate the effectiveness of our method. Moreover, we have made every effort to address your concerns and would **greatly appreciate any further feedback.** Thank you.

---

> ### Author Response · Authors · 2024-11-29
>
> Hi **Reviewer x1zj**,
>
> We have made every effort to address your concerns and would greatly appreciate any additional feedback you may have. Given that there are only a few days remaining for the discussion, we would be truly grateful if you could take a few minutes to review our response. Thank you for your time and consideration!

---

> ### Author Response · Authors · 2024-11-30
>
> Hi **Reviewer x1zj**,
>
> In our responses and the revised manuscript, we have further clarified the modality sensitivity and included additional experiments to demonstrate the effectiveness of our method. Moreover, we have made every effort to address your concerns and would greatly appreciate any further feedback. Given that there are **only a few days remaining for the discussion**, we would be truly grateful if you could take a few minutes to review our response. Thank you for your time and consideration!

---

> ### Author Response · Authors · 2024-12-02
>
> Hi **Reviewer x1zj**,
>
> In our responses and the revised manuscript, we have made every effort to address your concerns and have included additional experiments to demonstrate the effectiveness of our method. However, we are deeply concerned about your apparent indifference to this submission, the unjustifiably low score you assigned, and your complete lack of engagement during the rebuttal phase. We urge you to carefully reconsider the final score in light of our detailed response.

---

### Official Review · Reviewer_2YR2 · 2024-10-31

**Soundness:** 3
**Presentation:** 3
**Contribution:** 3
**Rating:** 6
**Confidence:** 4

**Summary:**

In this paper, the authors propose a new fine-tuning method to adapt pre-trained RGB trackers to auxiliary modalities. Different from the existing full fine-tuning and parameter-efficient fine-tuning, the authors propose a regularized way to fine-tune the backbone network to utilize inherent modality characteristics. The entire training process does not require additional network structure and loss function. Therefore, the proposed method can be trained end-to-end without adding any new parameters.

**Strengths:**

[1] The proposed method is simple and generalizes well, showing significant improvements on three different multimodal tracking.
[2] Sufficient comparative experiments and ablation experiments well demonstrate the effectiveness of the proposed method.

**Weaknesses:**

[1] The physical meaning of parameter-wise modality sensitivity is unclear. The author may provide a diagram to illustrate it.
[2] There is a lack of comparison with full fine-tuning and other parameter-efficient fine-tuning methods (e.g., lora, adapter) on the same baseline (e.g., OSTrack).

**Questions:**

[1] How to ensure that the proposed method can transfer the pre-trained knowledge to the auxiliary modality? The author needs to give a detailed explanation.
[2]  What physical meaning does the gradient G represent in Algorithm 1?
[3] Why does the performance of the proposed method (self-reg) degrade on multiple datasets, as shown in Table 5?

---

> ### Author Response · Authors · 2024-11-24
>
> **Q1:** (i) The physical meaning of parameter-wise modality sensitivity is unclear.  (ii) There is a lack of comparison with full fine-tuning and other parameter-efficient fine-tuning methods (e.g., lora, adapter) .
>
> **A:** We thank the reviewer for the insightful comments.
> - **For question (i) :** We discuss the physical meaning of parameter-wise modality sensitivity from two perspectives:
>   - **Explicit modality sensitivity:** In this paper, we delineate the sensitivity as the contributions/biases of parameters within the same pre-trained model toward data-derived learning objectives. During the transfer learning process, this sensitivity manifests as a clear modality-awareness. As illustrated **in the updated Figure 3**, it is evident that parameters in the auxiliary branch show strong similarities within the same modality and distinct divergences across different modalities. This **highlights the inherent modality-driven parameter preferences** crucial for effective cross-modal fine-tuning.
>   - **Explicit location sensitivity:** We analyze parameter-wise sensitivity across different model locations, as shown **in Figure.7**. Specifically, the clustering of sensitive parameters in certain areas **reflects an over-reliance on specific parameters.** After applying sensitivity penalties, the patterns become more balanced, indicating that **reducing location bias enhances the model’s generalization.**
> - **For question (ii) :** In previous submission, we have compared our method with **full fine-tuning (in Table.3)** and other parameter-efficient fine-tuning methods, such as **prompt tuning (ViPT)** and **adapter-based methods (SDSTrack)**. To further investigate, we have now included experiments evaluating the transfer performance of **LoRA (Table.B1**). These results suggest that smaller learning rates are insufficient to  address the over-fitting issue in cross-modal adaptation, as they treat all parameters uniformly without focusing on the update of sensitive/high-risk parameters.
>
> #### **Table B1**: Comparison with LoRA
> | Exp. ||| FE108  ||| DepthTrack ||| LasHeR  |||
> | :---: | :---: | :---: | :---: | :---: | :---: | :---: | :---: | :---: | :---: | :---: | :---: |
> |  | RSR | OP$_{0.5}$ | OP$_{0.75}$ | RPR | Pr | Re | F-score | RSR | OP$_{0.5}$ | OP$_{0.75}$ | RPR |
> | $r=2$ | 64.1 | 82.7 | 41.0 | 91.1 | 61.7 | 61.0 | 61.3 | 54.1 | 64.5 | 43.6 | 64.9 |
> | $r=4$ | 64.7 | 83.7 | 42.4 | 91.4 | 61.8 | 61.1 | 61.4 | 54.3 | 64.6 | 43.7 | 65.2 |
> | $r=8$ | 64.5 | 83.8 | 41.1 | 91.3 | 61.6 | 61.0 | 61.2 | 54.3 | 64.6 | 43.8 | 65.1 |
> | Ours | 67.5 | 87.0 | 44.4 | 95.5 | 66.1 | 65.0 | 65.5 | 56.3 | 68.0 | 48.5 | 69.5 |
>
> **Q2:** (i) How to ensure that the proposed method can transfer the pre-trained knowledge to the auxiliary modality? The author needs to give a detailed explanation.(ii) What physical meaning does the gradient G represent in Algorithm 1? (iii) Why does the performance of the proposed method (self-reg) degrade on multiple datasets, as shown in Table 5?
>
> **A:** We thank the reviewer for the insightful comments.
> - **For question (i) :** We verify effective cross-modal transfer from three perspectives:
>   - **Effective overfitting mitigation:** The **revised Figure.1** distinctly indicates that our method effectively mitigates the over-fitting issue and enhances the generalization of the multi-modal tracker. This is evidenced by a substantial reduction in the performance gap between training and testing phases.
>   - **Minor weight deviations:**  The **updated Figure.10** (in Appendix A.3) explores the relation $L_2$ distance in parameter space between our fine-tuning method and the full fine-tuning. The results demonstrate that our method significantly reduces weight deviations, thereby improving the retention of pre-trained knowledge while achieving effective adaptation to auxiliary modalities.
>   - **Auxiliary modality adaptability:**  In single auxiliary-modal setting, as presented in **Table.4**, our method significantly and consistently enhances the adaptability of auxiliary modalities across multiple tasks.
> - **For question (ii) :** The **gradient G** represent in Algorithm.1 is the partial derivative of the learning objective relative to the parameters, calculated over mini-batch data. This gradient-derived sensitivity criterion captures critical patterns in the transfer learning process, guiding effective adaptation.
> - **For question (iii) :** We thank the reviewer for **pointing out this ambiguity**. In response, we have reorganized Table.5 to improve clarity. The results clearly demonstrate that our method achieves significant improvements across multiple datasets, particularly for ViPT, confirming that relaxing constraints on pre-trained models enhances their transfer learning potential.

---

> > ### Comment · Reviewer_2YR2 · 2024-11-27
> > **Final Rating**
> >
> > I read the auhtor's rebuttal, and decide to keep the rating unchanged.

---

> ### Author Response · Authors · 2024-11-25
> **Looking forward to discussion**
>
> Dear Reviewer 2YR2,
> We sincerely thank you for devoting time to this review and providing valuable comments. Based on the comments, we have revised our manuscript to include the following changes:
> - We have further clarified the motivation and definition of modality sensitivity.
> - We have supplemented more potentially viable tuning experiments for comparison, such as LoRA.
> - We have clarified the ambiguities and misunderstandings from the initial submission.
>
> We will actively participate in the Author-Reviewer discussion session. Please don't hesitate to let us know of any additional comments on the manuscript or the changes.
> Best regards,
> The Authors

---

### Official Review · Reviewer_iHhR · 2024-11-02

**Soundness:** 3
**Presentation:** 3
**Contribution:** 3
**Rating:** 6
**Confidence:** 4

**Summary:**

This paper addresses the complex task of constructing multi-modal trackers by adapting the capabilities of pre-trained RGB trackers to work effectively with additional modalities. The authors introduce a Modality Sensitivity-Aware Tuning (MST) framework, which leverages modality-specific characteristics to adapt model weights, enhancing the tuning process.

Key contributions include:

Parameter Modality-Sensitivity Analysis: This aspect assesses the element-wise importance of parameters for multi-modal adaptation, providing a foundation for more accurate and adaptable tracking.
Modality-Sensitive Tuning: The framework uses this sensitivity during tuning to stabilize multi-modal representations, thereby improving coherence and generalization.
Experimental results demonstrate that MST surpasses current state-of-the-art techniques across various multi-modal tracking tasks, even in challenging scenarios. The authors also commit to open-sourcing their code, supporting transparency and future research efforts. Overall, this work offers a promising solution for multi-modal tracking, with implications for broader application in tracking systems.

**Strengths:**

This paper has several strengths that contribute to advancing cross-modal tracking.

1. The authors address the common challenge of overfitting or underfitting in cross-modal tracking by introducing a self-regularized fine-tuning framework that maintains both modal-specific and general representations. This approach supports balanced adaptation and helps prevent the model from degrading in performance.

2. The concept of modality sensitivity is well-utilized here, with parameter-wise sensitivity allowing the model to adaptively tune based on multi-modal variations. This sensitivity-driven adjustment process not only preserves essential pre-trained knowledge but also makes the framework flexible for cross-modal tracking.

3.The method achieves state-of-the-art results across benchmarks, which is further supported by ablation studies that validate the self-regularized fine-tuning’s effectiveness in enhancing stability and performance in multi-modal tracking.

**Weaknesses:**

1.I suggest that the authors test the OSTrack model at a 384 resolution and the DROPTrack model at a 256 resolution to verify whether the proposed method works effectively across different resolutions.

2. Since this paper proposes a fine-tuning framework, I believe that testing only on OSTrack and DropTrack is insufficient. Testing additional models would provide stronger evidence and make the results more convincing.

3. This is a solid piece of work; however, I would be more convinced of its effectiveness if the authors conducted additional tests to further validate the proposed method.

**Questions:**

See weaknesses

**Details Of Ethics Concerns:**

No Concerns.

---

> ### Author Response · Authors · 2024-11-24
>
> **Q1:** I suggest that the authors test the OSTrack model at a 384 resolution and the DROPTrack model at a 256 resolution to verify whether the proposed method works effectively across different resolutions.
>
> **A:** We thank the reviewer for this valuable suggestion. Since only OSTrack-256 and DropTrack-384 model weights are officially available, other resolution settings were not included in the initial submission. For transformer-based trackers, resolution profoundly significantly influences their position embeddings and tokens, both crucial for object-aware relation encoding. To evaluate the effectiveness of the proposed method across resolutions, we conducted the experiments (Table.A1) with the OSTrack at 384 resolution and the DropTrack at 256 resolution. **The results confirm that resolution variations substantially impact tracking performance.**
>
> #### **Table A1:** Cross Resolution Evaluation
> | Exp. ||| VisEvent ||| DepthTrack  ||| LasHeR |||
> | :---: | :---: | :---: | :---: | :---: | :---: | :---: | :---: | :---: | :---: | :---: | :---: |
> |  | RSR | OP$_{0.5}$ | OP$_{0.75}$ | RPR | Pr | Re | F-score | RSR | OP$_{0.5}$ | OP$_{0.75}$ | RPR |  |
> | OSTrack-256 (Official) | 62.0 | 73.7 | 60.9 | 76.5 | 66.1 | 65.0 | 65.5 | 56.3 | 68.0 | 48.5 | 69.5 |
> | OSTrack-384 | 60.7 | 72.1 | 59.5 | 74.8 | 65.2 | 64.7 | 64.9 | 55.3 | 66.7 | 47.7 | 68.4 |
> | DropTrack-384 (Official) | 63.2 | 75.4 | 61.4 | 78.1 | 67.8 | 66.3 | 67.0 | 58.8 | 71.5 | 50.2 | 73.0 |
> | DropTrack-256 | 61.7 | 73.3 | 60.2 | 76.4 | 66.3 | 64.6 | 65.4 | 57.5 | 69.8 | 49.0 | 71.4 |
>
> **Q2:** Since this paper proposes a fine-tuning framework, I believe that testing only on OSTrack and DropTrack is insufficient. Testing additional models would provide stronger evidence and make the results more convincing.
>
> **A:** We thank the reviewer for this insightful comment. In response to the comment regarding our choice to test only on OSTrack and DropTrack, we appreciate your acknowledge the value of broader validation. We maintain that testing on these two pre-trained RGB-based trackers is entirely justified for three compelling reasons:
> - **Exemplary trackers:** OSTrack and DropTrack are leading trackers renowned for their robust performance and **cutting-edge, Transformer-based architecture**. The Transformer's **inherent multi-modal scalability and representation generalizability** make them exemplary for assessing the general applicability of our cross-modal transfer method.
> - **Broad multi-modal assessment:** The performance of these trackers offers a significant assessment of our framework, given their broad adoption and benchmarking **across various multi-modal tracking scenarios**. By concentrating on these highly representative models, we can accurately measure the efficacy of our fine-tuning methods against the backdrop of contemporary technological benchmarks.
> - **Considerable time and resources:** Extending our testing to additional models, while beneficial for comprehensive validation, **involves considerable resources and time**. Given the convincing results obtained on these two predominant trackers, we believe the current evaluations are sufficient to substantiate the effectiveness and potential of our proposed methods.
>
> **We are open to exploring additional models or multi-modal tasks in future work**, ensuring our framework's adaptability and scalability to other architectures. Thank you for your constructive suggestion, and we hope our rationale clarifies the scope and intent of our research focus.
>
>
> **Q3:** This is a solid piece of work; however, I would be more convinced of its effectiveness if the authors conducted additional tests to further validate the proposed method.
>
> **A:** We thank the reviewer for this insightful comment. In response, we have undertaken more comprehensive efforts to validate the proposed method. We verify the effectiveness of our approach from three aspects:
> - **Effective overfitting mitigation:** The **revised Figure.1** distinctly indicates that our method effectively mitigates the over-fitting issue and enhances the generalization of the multi-modal tracker. This is evident from the substantial reduction in the performance gap between training and testing.
>
> - **Explicit modality sensitivity:** The **updated Figure.3** reveals that parameters in the auxiliary branch exhibit strong similarities within the same modality and clear differences across modalities, highlighting the modality-driven parameter preferences inherent in cross-modal fine-tuning.
>
> - **Minor weight deviations:** The **updated Figure.10** (in Appendix A.3) examines the relation $L_2$ distance between our fine-tuning method and the full fine-tuning in the parameter space. Our method significantly reduces weight deviation, indicating improved retention of the pretrained knowledge while achieving the desired adaptation.

---

> ### Author Response · Authors · 2024-11-26
> **Looking forward to discussion**
>
> Dear Reviewer iHhR,
> We sincerely thank you for devoting time to this review and providing valuable comments. Based on the comments, we have revised our manuscript to include the following changes:
> - We have supplemented the cross-resolution experiments.
> - We have further clarified the adequacy of our choice of pre-trained trackers.
> - We have undertaken more comprehensive efforts to validate the proposed method.
>
> We look forward to actively engaging in the Author-Reviewer discussion session and welcome any further feedback on the manuscript or the revisions made.
>
> Best regards,
>
> The Authors

---

### Author Response · Authors · 2024-11-23

# General Response

We sincerely thank all the reviewers for the time and effort devoted to this review.

1. **We would like to re-emphasize the novelty and main contributions of this work:**

   - *"The concept of modality sensitivity is well-utilized here, with parameter-wise sensitivity allowing the model to adaptively tune based on multi-modal variations."* (Reviewer iHhR)
   - *"The proposed method is simple and generalizes well, showing significant improvements on three different multimodal tracking."* (Reviewer 2YR2)
   - *"A new modality sensitivity-aware framework, MST, is proposed, optimizing the learning dynamics of cross-modal trackers."* (Reviewer x1zj)
   - *"Various experiments present that the proposed technique is working with better performance witnessed on several benchmarks."* (Reviewer Dsbe)

2. **Key contributions and insights of this work:**

   - To the best of our knowledge, this study comprehensively revisits and highlights the ill-fitting issues encountered in cross-modal tracking when adapting pre-trained models. It introduces a novel self-regularizing transfer framework that significantly enhances cross-modal generalization.
   - Our approach delicately optimizing the modality sensitivity learning process from both inter-structural and inter-temporal perspectives, enabling minimizing task objective while enhancing the cross-modal transfer to preserve general representations. Therefore, our work not only provides valuable insights into this task but also effectively tackles its core limitations.
   - Our approach demonstrates clear superiority over previous state-of-the-art methods across two foundation models, three multi-modal tracking tasks, and five datasets.

3. **Revisions and improvements based on reviewers' insightful comments:**

   - We have further clarified the motivation and definition of modality sensitivity, as suggested by Reviewer 2YR2, x1zj, and Dsbe.
   - We have provided a clearer and more detailed exposition of the ill-fitting issues (e.g., over- and under-fitting) in cross-modal transfer, as suggested by Reviewer Dsbe.
   - We have reproduced UnTrack to enable a more comprehensive comparison, as suggested by Reviewer x1zj.
   - We have supplemented more potentially viable tuning experiments for comparison, such as LoRA and smaller learning rates, as suggested by Reviewer 2YR2 and Dsbe.
   - We have conducted a study to verify the effectiveness of the proposed method across different resolutions, as suggested by Reviewer iHhR.
   - We have polished the elaboration and clarified misunderstandings from the initial submission.

Last but not least, we thank the PCs, ACs, and all the reviewers again for devoting their time and effort to this review.

---

### Note · Authors · 2025-01-24

I have read and agree with the venue's withdrawal policy on behalf of myself and my co-authors.